# Tuning ubiquitin transfer by RING E3 ubiquitin ligases through the linchpin residue

Mark A Nakasone[1,*], Lori Buetow[1,*], Mads Gabrielsen[1,*], Syed F Ahmed[1], Karolina A Majorek[1], Gary J Sibbet[1], Brian O Smith[2], Danny T Huang[1,3]

RING family ubiquitin ligases (E3s) employ the RING domain to recruit the E2 thioester ubiquitin (E2~Ub) intermediate to catalyze the transfer of ubiquitin (Ub) to substrates. A cationic Arg linchpin (LP) residue in the RING domain plays a key role in stabilizing the interface with E2~Ub, but the identity of the LP residue varies across E3s. Here, we investigate how the LP residue contributes to ubiquitination. Using the model RNF38 system, we demonstrate that substitution of LP[Arg] to the other 19 available amino acids modulates ubiquitination, ranging from minor reduction to complete abolition. The identity of the LP residue influences E2~Ub binding but does not correlate with E3 activity. NMR and X-ray crystallography analyses reveal that RNF38 LP[Arg] variants stabilize E2~Ub in a catalytically competent conformation to varying degrees. By altering the LP residue in XIAP, we show that the XIAP[Y485R] variant promotes E2~Ub stabilization and enhances substrate ubiquitination in cells. Our work demonstrates the importance of the LP residue in modulating E2~Ub conformation to control ubiquitination.

## Introduction

In eukaryotes, most intracellular proteins are posttranslationally modified with the 76-residue protein, ubiquitin (Ub) (Hershko & Ciechanover, 1998). The covalent attachment of Ub to substrate proteins is achieved through the E1, E2, E3 cascade (Buetow & Huang, 2016). In short, an E1-activating enzyme forms a thioester intermediate with the C-terminal Gly[76] of Ub, and Ub is then transferred to the active-site Cys on one of ~35 (human) E2 ubiquitin–conjugating enzymes, resulting in another thioester intermediate, E2~Ub[1]. Several families of E3 ubiquitin ligases can then recruit E2~Ub for attachment of the C-terminal Gly[76] of Ub to primary amines of substrates (Deshaies & Joazeiro, 2009). The ~630 E3 ubiquitin ligases that modify the human proteome recruit

both their subset of substrates and E2~Ub; however, Ub transfer occurs by distinct mechanisms depending on the E3 family (Li et al, 2008).

Really Interesting New Gene (RING) family E3s are wide-ranging regulators of physiology and the largest family of E3 ligases. First described in 1991, RING domains have since been understood to catalyze Ub transfer by positioning E2~Ub in proximity of a nucleophile on the substrate (Freemont et al, 1991; Deshaies & Joazeiro, 2009). In addition to the protein substrate and the E3 itself, any of eight sites (Met[1], Lys[6], Lys[11], Lys[27], Lys[29], Lys[33], Lys[48], and Lys[63]) from a previously attached Ub can accept Ub donor (Ub[D]) from E2~Ub (Komander & Rape, 2012). Such Ub-Ub linkages enable the formation of a wide variety of differently linked polymeric chains of Ub (Varadan et al, 2002; Nakasone et al, 2013). In most RING E3s, the RING domain is found at the N or C terminus; however, some E3 subfamilies, including the CBL family, incorporate the RING domain internally. Aside from the RING domain, RING E3s at a minimum possess one or more substrate binding domains or a short sequence that enables association with larger complexes, as is the case with APC11 and RBX1 (ROC1/RNF75) (Furukawa et al, 2000; Buetow & Huang, 2016; Yamaguchi et al, 2016). All RING domains have a set of conserved Cys and His residues that coordinate two structural Zn[2+] ions (Metzger et al, 2014). In addition to their diverse domain architecture, RING E3 function depends on the multimerization state of the RING—monomer, homo- or heterodimer, or within large multi-subunit complexes—which is often key in regulating assembly (Metzger et al, 2014; Buetow & Huang, 2016).

The N-terminal helix of the ubiquitin-conjugating (UBC) fold of E2s is the conserved binding site for canonical RING domains. Affinity of RING E3s for E2~Ub[D] is greatly enhanced over E2 alone because of additional interactions between the RING domain and Ub[D] within the ternary complex (Dou et al, 2012b; Plechanovová et al, 2012; Buetow et al, 2015). Therefore, catalysis can be impaired by mutating residues involved in contacting the RING domain and either E2, Ub[D], or both. In E2~Ub[D] complexes with RING E3s, Ub[D] is

[1]Cancer Research UK Scotland Institute, Glasgow, UK    [2]School of Molecular Biosciences, University of Glasgow, Glasgow, UK    [3]School of Cancer Sciences, University of Glasgow, Glasgow, UK

Correspondence: mark.nakasone@glasgow.ac.uk; d.huang@crukscotlandinstitute.ac.uk
Mads Gabrielsen's present address is Neil Bulleid Integrated Protein Analysis, University of Glasgow, Glasgow, UK
*Mark A Nakasone, Lori Buetow, and Mads Gabrielsen contributed equally to this work

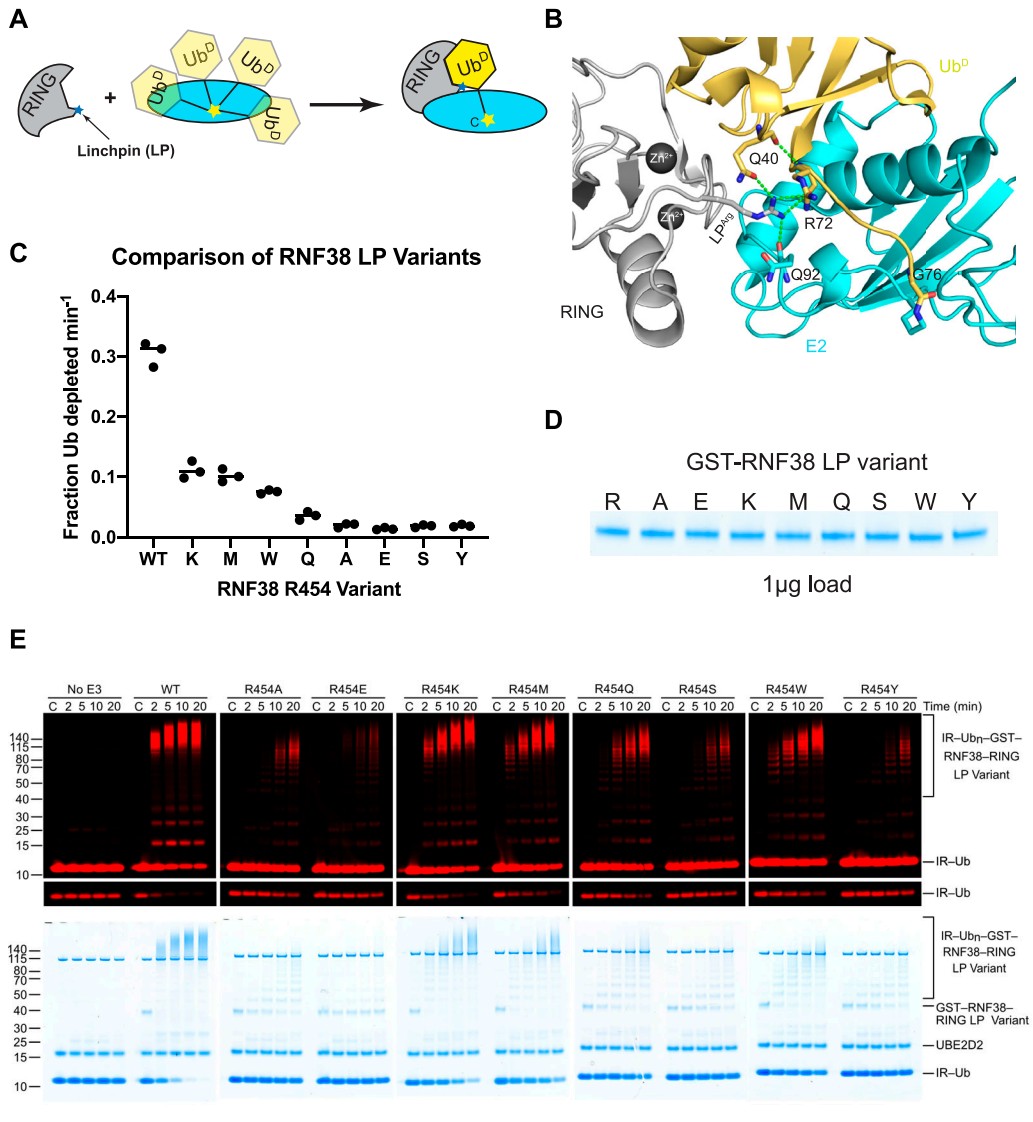

**Figure 1. LP residue in RING E3s is critical for ubiquitination.**
**(A)** Schematic of a RING domain (gray) stabilizing Ub[D] (yellow) conjugated to E2 (cyan) in the primed conformation. LP[Arg] is denoted as a blue star and E2 catalytic Cys as a yellow star. **(B)** Cartoon representation of the primed conformation in the canonical RNF38-UBE2D2–Ub[D] system (PDB 4V3K). LP[Arg] (shown as sticks) forms key contacts with residues in Ub[D] to maintain the primed conformation. $Zn^{2+}$ ions are shown as dark gray spheres. **(C)** Fraction of depleted mono-Ub was quantified for the indicated RNF38 LP variants. Individual data are plotted from three independent experiments (n = 3). **(C)** IR–Ub in (C) indicates fluorescence-labeled Ub. **(D)** Standard 1 μg load of GST-RNF38 LP variants from (D). **(E)** Detection of near IR (680 nm)–labeled Ub (IR-Ub) on reduced SDS–PAGE across an autoubiquitination time course with GST-RNF38 LP variants. A lower exposure of IR-Ub bands was shown for quantification. Coomassie staining is shown at the bottom.

presented in a closed conformation where the thioester bond between Gly[76] of Ub[D] and the active-site Cys of E2 is primed for optimal nucleophilic attack (Metzger et al, 2014; Buetow & Huang, 2016). Notably, the position of a cationic Arg residue in the RING domain is conserved in most structures of RING E3/E2~Ub[D] complexes (Dou et al, 2012b; Plechanovová et al, 2012; Dou et al, 2013; Branigan et al, 2015; Koliopoulos et al, 2016; Dawidziak et al, 2017; Nomura et al, 2017; Nayak & Sivaraman, 2018; Kiss et al, 2019; Patel et al, 2019; Middleton et al, 2020; Nakasone et al, 2022; Paluda et al, 2022), where the Arg side chain forms a network of hydrogen bonds with Ub[D] and E2 to stabilize the closed E2~Ub[D] conformation (Fig 1A). The importance of the Arg residue for catalysis has led to it

being named the "linchpin" (LP) (Pruneda et al, 2012). Collectively, studies demonstrate that RING E3 binding induces a shift in the E2~Ub[D] conformation from an open conformation to a closed conformation to increase Ub transfer activity (Fig 1B) (Dou et al, 2012b; Pruneda et al, 2012; Branigan et al, 2020). Alteration in the LP Arg residue influences this equilibrium and reduces Ub transfer activity (Dou et al, 2012b; Plechanovová et al, 2012; Pruneda et al, 2012; Buetow et al, 2015; Kiss et al, 2019; Lips et al, 2020). It is interesting to note that not all RING E3s have a cationic residue at the LP position (Lips et al, 2020). Recent work by Lips and coworkers highlights the impact of the LP residue in the yeast ERAD system, where two RING E3 ligases, Hrd1 and Doa10, function differently

with their two E2 enzyme partners, Ubc6 and Ubc7 (Lips et al, 2020). Doa1 has a His residue in the LP position and binds both E2~Ub–conjugated complexes with low binding affinity (200–700 $\mu$M) (Lips et al, 2020). In addition, Ubc6 has a high basal activity, which is not reliant on the presence of an Arg in the linchpin position, and is used by Doa1 as a "priming" E2, whereas Ubc7 is used as the "elongating" E2. In contrast, Hrd1, which has Arg in its linchpin position, has a high binding affinity to Ubc7~Ub and uses the same E2 for both "priming" and "elongating" (Lips et al, 2020). This observation prompted us to investigate how the LP residue affects ubiquitination through an in-depth comparison of well-studied RING E3s.

In this work, we describe the role of the conserved LP residue in catalysis and how varying the identity of this single residue can be used to selectively enhance or abolish E3 ligase activity. We chose the RING domain of RNF38 to use as a model monomeric RING with a native LP$^{Arg}$. Through mutagenesis of the LP residue and subsequent functional and biophysical measurements, we demonstrate that RING/E2~Ub$^D$ binding does not directly correlate with E3 ligase activity. Analysis of Ub$^D$ conformation by solution nuclear magnetic resonance (NMR) spectroscopy and by determining a crystal structure of RNF38 LP$^{Tyr}$ in complex with E2~Ub$^D$ reveals varying degrees of Ub$^D$ stabilization by different LP residues. These results indicate the importance of E2~Ub$^D$ conformation in controlling E3 activity, with Arg being the most effective LP residue. Making a LP Y485R substitution in XIAP enhances the stabilization of closed E2~Ub$^D$ and E3 activity in vitro and when expressed in HEK293 cells. Together, this work provides a broad understanding of how the LP residue modulates RING E3–dependent ubiquitination.

# Results

### A critical basic residue is conserved in RING E3s

RING E3s employ a cationic LP residue to position Ub$^D$ on E2~Ub for catalysis (Dou et al, 2012b; Plechanovová et al, 2012; Pruneda et al, 2012) (Figs 1A and B and S1). The consensus of RING E3/E2~Ub$^D$ structures demonstrates that the LP residue is positioned at the interface between Ub$^D$ and E2 to stabilize the ternary complex (Fig S1B–G). In addition, sequence alignment of the canonical RING domain from RNF38 to the human genome shows a high incidence of a cationic residue at the LP position, with Arg being the most represented (Lips et al, 2020). However, given the variability observed in the identities of the LP residues, its effects on E2~Ub binding, the kinetics of Ub transfer, and the ability to induce a catalytically competent E2~Ub$^D$ conformation have yet to be fully explored. To elucidate the role of the LP residue in these key functions, we chose the RING domain of RNF38 (RNF38$^{RING}$) as a model system as it is monomeric and readily crystallizes with active-site Cys-to-Lys variants of the E2s: UBE2D2–Ub and UBE2K–Ub that present Ub$^D$ as stable isopeptide linkage (Buetow et al, 2015; Nakasone et al, 2022).

### RNF38 LP mutants alter binding to UBE2D2 and UBE2D2−Ub

The RNF38 LP residue, Arg$^{454}$, was mutated to each of the other 19 naturally occurring amino acids, and their binding affinities for UBE2D2 alone and UBE2D2−Ub were determined using surface plasmon resonance (SPR) (Fig S2). The UBE2D2$^{S22R}$ mutant was used to abrogate any effect caused by noncovalent binding of Ub to the backside region of the E2, and UBE2D2$^{S22R,C85K}$ was used for conjugation to Ub$^D$ (henceforth referred to as UBE2D2$^{S22R,C85K}$–Ub) (Sakata et al, 2010; Buetow et al, 2015). The binding affinities of UBE2D2$^{S22R}$ and UBE2D2$^{S22R,C85K}$–Ub to RNF38$^{RING}$-WT were measured to be $K_d$ = 105 ± 2 and 2.5 ± 0.4 $\mu$M, respectively (Table 1). Most substitutions of Arg$^{454}$ resulted in less than a twofold change to the binding of UBE2D2$^{S22R}$ alone, with RNF38$^{RING}$-R454W displaying a slight increase in UBE2D2$^{S22R}$ binding. All RNF38$^{RING}$ LP variants exhibited stronger affinity for UBE2D2$^{S22R,C85K}$–Ub binding than for UBE2D2$^{S22R}$ alone. RNF38$^{RING}$-WT displayed ~40-fold higher affinity for UBE2D2$^{S22R,C85K}$–Ub compared with UBE2D2$^{S22R}$ in agreement with prior studies (Buetow et al, 2015). In contrast, all other RNF38$^{RING}$ LP variants only showed a two- to sevenfold enhancement for UBE2D2$^{S22R,C85K}$–Ub binding compared with UBE2D2$^{S22R}$. Substitution of Arg$^{454}$ to Trp (W) or Tyr (Y) resulted in a ~9.5-fold reduction in UBE2D2$^{S22R,C85K}$–Ub binding affinity compared with WT, whereas the charge-swapped R454E (Glu) substitution severely disrupted the binding affinities to UBE2D2$^{S22R}$ and UBE2D2$^{S22R,C85K}$–Ub (Table 1).

### LP variants differentially modulate ubiquitination efficiency

We selected a number of RNF38 LP variants (R454K, R454M, R454Q, R454Y, R454W) whose binding affinities to UBE2D2$^{S22R,C85K}$–Ub are around 10-fold lower than WT and assessed their impact on ubiquitination. In addition, we selected RNF38 LP variants R454S and R454A based on the chemical properties of their side chains and the natural occurrence of LP$^{Ala}$ in some human RING domains. The severe abrogation of binding by the R454E mutant was also investigated. For our functional assay, UBE2D2 was precharged with Ub$^D$ to form UBE2D2–Ub and GST-RNF38$^{RING}$ LP variants were added to initiate the autoubiquitination reaction (Fig 1C–E) allowing the fraction of mono-Ub depleted per minute to be determined. At ~33% of mono-Ub depleted per minute, the WT LP$^{Arg}$ in RNF38$^{RING}$ is the most efficient followed by the R454K and R454M variants, whereas all other LP variants showed minimal activity (Figs 1C–E and S3A and B). The R454E variant with its severe reduction in UBE2D2$^{S22R,C85K}$–Ub binding affinity also showed only trace activity. The deficiency of the R454Y variant in the ubiquitination assay compared with the R454K, R454M, and R454W variants was unexpected as it exhibited the second-best binding affinity to UBE2D2$^{S22R,C85K}$–Ub after RNF38$^{RING}$-WT (Table 1). To address this, we set out to elucidate how the LP residue in RNF38 impacts the coordination of Ub$^D$ in E2~Ub.

### Structure of free RNF38 suggests rigidity in RING domains

RNF38$^{RING}$ has been crystallized in complex with UBE2D2–Ub previously (Buetow et al, 2015), yet no high-resolution crystal structure exists of the RNF38$^{RING}$ in isolation. To establish whether

**Table 1. Surface plasmon resonance analysis of RNF38 LP variants binding UBE2D2[S22R] and UBE2D2[S22R,C85K]–Ub.**

| Ligand GST-tagged RNF38 | UBE2D2[S22R] Kd ($\mu$M) | UBE2D2[S22R,C85K]–Ub Kd ($\mu$M) | Fold difference for UBE2D2[S22R] | Fold difference for UBE2D2[S22R,C85K]–Ub |
|---|---|---|---|---|
| WT | 105 ± 2 | 2.5 ± 0.4 | | |
| R454W | 63.1 ± 0.8 | 24.3 ± 0.8 | 0.60 | 9.6 |
| R454Q | 101 ± 5 | 29.2 ± 3.6 | 0.96 | 11.6 |
| R454I | 111 ± 2 | 42.0 ± 0.1 | 1.06 | 16.7 |
| R454K | 132 ± 3 | 30.2 ± 3.7 | 1.26 | 12.0 |
| R454Y | 138 ± 3 | 23.8 ± 1.1 | 1.32 | 9.4 |
| R454M | 139 ± 2 | 31.2 ± 1.7 | 1.33 | 12.4 |
| R454S | 153 ± 5 | 59.5 ± 2.1 | 1.46 | 23.6 |
| R454T | 160 ± 4 | 63.0 ± 1.8 | 1.52 | 25.0 |
| R454L | 166 ± 7 | 46.7 ± 1.1 | 1.58 | 18.5 |
| R454A | 167 ± 6 | 60.4 ± 1.5 | 1.59 | 24.0 |
| R454V | 173 ± 5 | 53.9 ± 1.4 | 1.65 | 21.4 |
| R454F | 173 ± 5 | 45.4 ± 1.4 | 1.65 | 18.0 |
| R454H | 200 ± 11 | 55.7 ± 4.2 | 1.91 | 22.1 |
| R454C | 209 ± 9 | 56.5 ± 1.1 | 1.99 | 22.4 |
| R454P | 245 ± 12 | 82.7 ± 3.1 | 2.34 | 32.8 |
| R454D | 270 ± 16 | 69.8 ± 2.8 | 2.57 | 27.7 |
| R454G | 393 ± 21 | 91 ± 3 | 3.75 | 36.1 |
| R454N | 623 ± 77 | 106 ± 3 | 5.94 | 42.0 |
| R454E | 755 ± 170 | 111 ± 6 | 7.20 | 44.0 |

SEM is indicated. The number of replicates, representative sensorgrams, and binding curves are shown in Fig S2.

Arg[454], the native LP residue in RNF38, is constrained by other residues in the absence of E2~Ub, we determined the structure of RNF38[RING] to 1.2 Å (Fig 2A and Table 2). RNF38[RING] adopts a similar structure to that seen in the RNF38[RING]-UBE2D2–Ub complex (r.m.s. deviation of 0.58 Å for 64 C$\alpha$ atoms). The location of LP[Arg] within the RNF38 RING domain remains unchanged, with its side chain adopting a slightly different orientation, most likely as a consequence of crystal packing (Fig S4).

### Structure of E2~Ub with RNF38[RING] LP variants perturbs donor Ub

Having determined the crystal structure of RNF38[RING] alone to elucidate the position of Arg[454], we subsequently attempted to crystallize a subset of RNF38[RING] LP variants, namely, R454Y, R454K, R454D, R454N, R454W, and R454M in complex with UBE2D2[S22R,C85K]–Ub. The RNF38[RING]-R454Y/UBE2D2[S22R,C85K]–Ub complex diffracted to 2.6 Å (Fig 2B and Table 2). Crystals of UBE2D2[S22R,C85K]–Ub with RNF38[RING]-R454W, RNF38[RING]-R454K, and RNF38[RING]-R454M were obtained, but diffraction was too poor for structural refinement. In the RNF38[RING]-R454Y/UBE2D2[S22R,C85K]–Ub complex, RNF38[RING]-R454Y/UBE2D2 interaction is preserved as observed in the prior structure of the RNF38[RING]/UBE2D2[S22R,C85K]–Ub complex (Fig S5). In contrast, Ub[D] is not in the closed conformation but rather oriented away from RNF38[RING]-R454Y (Figs 2B and S5A and B). Analysis of crystal contacts showed that the Arg[42] surface of Ub[D] packs against the Tyr[399], Gln[407], and

Glu[434] regions of RNF38 in the asymmetric unit (Fig S6A). To explore the potential noncovalent RNF38/Ub interaction in the absence of E2, we titrated monomeric Ub([15]N) to a 1:1 M ratio with RNF38[RING] (Fig S6B and C), but did not observe binding. Therefore, this could represent an extremely weak site for Ub with $K_d$ in the hundreds of micromolar range or higher and the contacts observed in the crystals are likely due to crystal packing akin to those reported for RNF38[RING]/UBE2K–Ub/K48-Ub$_2$ (Nakasone et al, 2022).

### NMR reveals LP arginine is essential for priming Ub[D]

As the orientation of Ub[D] in E2~Ub is critical for efficient transfer, we used solution NMR to directly observe how Ub[D] within E2~Ub bound to a selection of LP variants. For this purpose, [15]N-labeled Ub[D] was conjugated to the active-site C85K in UBE2D2[S22R,C85K] via an isopeptide bond. Notably, Ub[D] in the related UBE2D3–Ub system was reported to be unconstrained and conformationally flexible (Pruneda et al, 2011). Comparison of the UBE2D2[S22R,C85K]–Ub([15]N) [1]H,[15]N-HSQC spectrum with that of monomeric Ub([15]N) confirmed similar behavior in Ub[D] (Fig S7A–D). Upon reaching a 1:1 M ratio in a titration of RNF38[RING]-WT with UBE2D2[S22R,C85K]–Ub([15]N), most of the signals from Ub[D] were attenuated in the [1]H,[15]N-HSQC spectrum, indicating that they had entered the intermediate exchange regime, whereas many of the remaining signals displayed substantial chemical shift perturbations (CSPs) (Figs 2C and S7). The changes in CSPs are

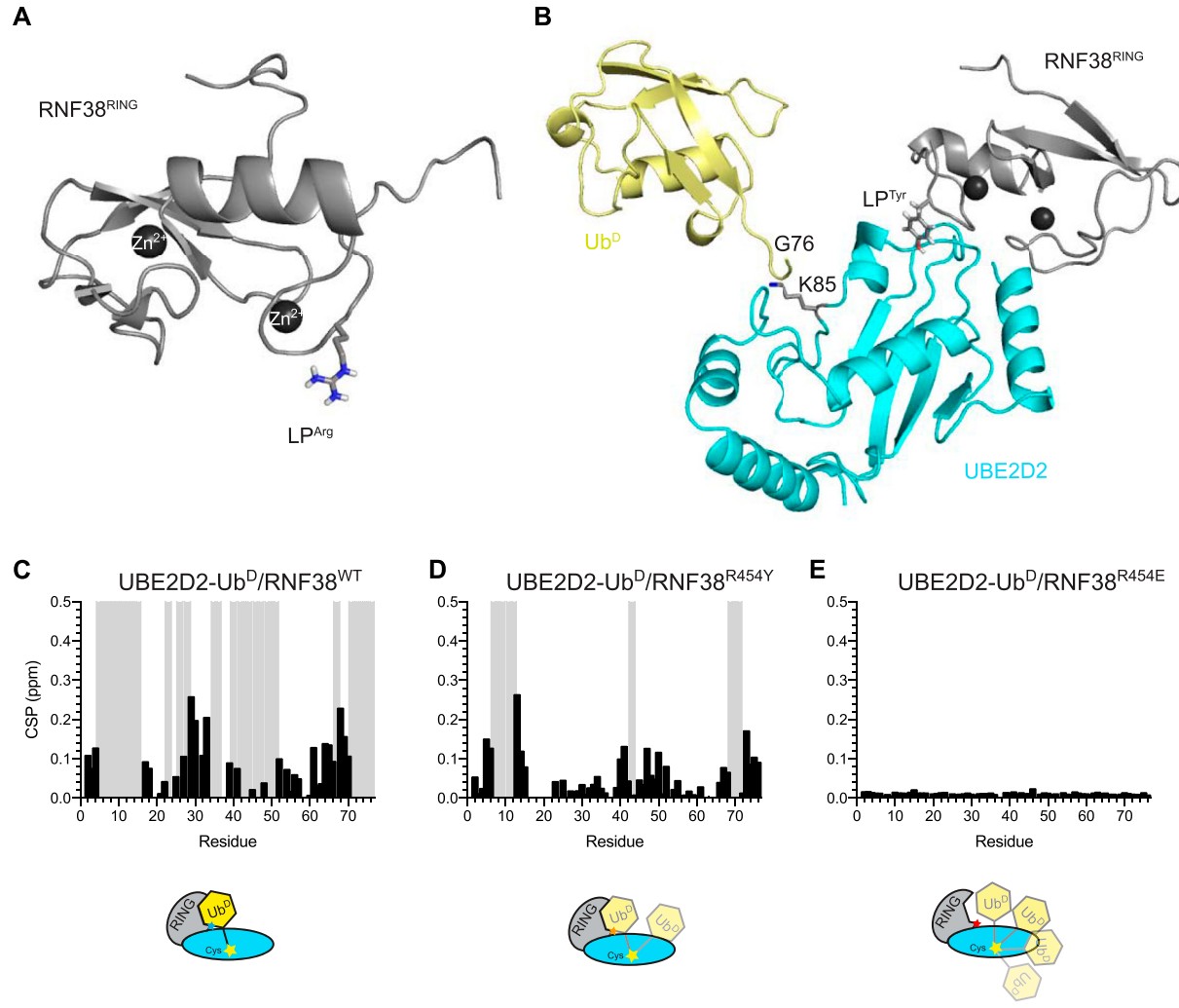

**Figure 2. Structure of UBE2D2–Ub$^D$ and RNF38 LP variants.**
**(A)** Cartoon representation of RNF38$^{RING}$ domain in isolation with LP Arg$^{454}$ shown as sticks. **(B)** Cartoon representation of the RNF38$^{R454Y}$/UBE2D2$^{S22R,C85K}$–Ub complex with RNF38$^{RING}$ domain (gray) with Zn$^{2+}$ ions (dark gray spheres), Ub$^D$ (yellow), UBE2D2 (cyan). **(C, D, E)** Residue-specific chemical shift perturbations (black) and signal attenuations (gray) for UBE2D2$^{S22R,C85K}$–Ub$^D$ ($^{15}$N) combined with (C) RNF38$^{WT}$, (D) RNF38$^{R454Y}$, and (E) RNF38$^{R454E}$ at a 1:1 M ratio.

consistent with RNF38$^{RING}$ binding directly to Ub$^D$, in agreement with the structure of the RNF38$^{RING}$/UBE2D2–Ub complex (Buetow et al, 2015), where Ub$^D$ is constrained in a closed conformation. Analysis of CSPs and signal attenuations provided us with a sensitive approach to monitor Ub$^D$ conformation upon binding to RNF38$^{RING}$. We next tested interaction with RNF38$^{RING}$-R454Y, which produced smaller CSPs and attenuated the Ub($^{15}$N) signals less than RNF38$^{RING}$-WT (Fig 2D), suggesting that RNF38$^{RING}$-R454Y may not be able to bind Ub$^D$ and restrict it into the closed conformation, as observed with RNF38$^{RING}$-WT. To highlight the importance of LP$^{Arg}$, RNF38$^{RING}$-R454E was titrated into UBE2D2$^{S22R,C85K}$–Ub($^{15}$N), which resulted in virtually no changes between $^1$H,$^{15}$N-HSQC spectra, suggesting Ub$^D$ remained unconstrained (Figs 2E and S7). Together, these three LP variants demonstrate how the identity of LP contributes to maintaining the closed conformation of Ub$^D$. Furthermore, this NMR assay is in agreement with the weak binding and deficient

ubiquitination we observed for RNF38$^{RING}$-R454E. Therefore, the reduced activity of RNF38$^{RING}$-R454Y in ubiquitination is explained by the suboptimal positioning of Ub$^D$, as revealed by both NMR and crystallography.

### XIAP-RNF38 chimera reveals necessity of LP for substrate ubiquitination

The exact substrates and cellular function of RNF38, as well as many other E3 ligases, are poorly defined. Therefore, to determine the impact of LP variants in the context of a known E3/substrate pair, we engineered an XIAP-RNF38 chimera model system (Fig 3A). In this system, the XIAP substrate–recruiting BIR2 and BIR3 domains are retained at the N terminus to bind substrate SMAC (Srinivasula et al, 2001), whereas the C-terminal RING domain is substituted with RNF38$^{RING}$ to yield XIAP$^{145-430}$-RNF38$^{389-C}$ (referred to as XIAP-RNF38 from hereon). A similar

**Table 2.  X-ray crystallography data collection and refinement statistics: structures for WT RNF38[RING] and RNF38[R454Y]/UBE2D2[S22R,C85K]–Ub.**

| | RNF38 | RNF38[RY]-UBE2D2–Ub |
|---|---|---|
| PDB | 9Q88 | 9Q8Y |
| Wavelength | 0.9795 | 0.976250 |
| Resolution range | 31.62–1.2 (1.28–1.2) | 43.92–2.627 (2.77–2.63) |
| Space group | $P\,2_1\,2_1\,2_1$ | $P\,6_2$ |
| Unit cell | | |
| a, b, c | 31.82, 39.309, 53.208 | 116.079, 116.079, 90.315 |
| α, β, γ | 90, 90, 90 | 90, 90, 120 |
| Total reflections | 91,996 (224) | 205,854 (8,387) |
| Unique reflections | 17,218 (184) | 20,539 (879) |
| Multiplicity | 5.3 (1.2) | 10.0 (9.5) |
| Completeness (%) | 80.04 (15.04) | 99.0 (85.3) |
| Mean I/sigma(I) | 18.1 (1.9) | 22.9 (2.3) |
| Wilson B-factor | 10.24 | 79.94 |
| R-merge | 0.053 (0.270) | 0.052 (0.856) |
| R-pim | 0.026 (0.510) | 0.020 (0.308) |
| CC1/2 | 0.999 (0.937) | 0.999 (0.853) |
| Reflections used in refinement | 17,205 (848) | 20,506 (2,781) |
| Reflections used for R-free | 809 (38) | 995 (139) |
| R-work | 0.1512 (0.2590) | 0.2317 (0.3709) |
| R-free | 0.1684 (0.2509) | 0.2746 (0.4265) |
| Number of nonhydrogen atoms | 741 | 4,253 |
| Protein atoms | 644 | 4,249 |
| Ligands | 2 | 4 |
| Solvent | 95 | 0 |
| Protein residues | 79 | 553 |
| RMS(bonds) | 0.007 | 0.013 |
| RMS(angles) | 1.04 | 1.77 |
| Ramachandran favored (%) | 94.81 | 91.47 |
| Ramachandran allowed (%) | 5.19 | 7.82 |
| Ramachandran outliers (%) | 0.00 | 0.71 |
| Rotamer outliers (%) | 0.00 | 0.66 |
| Clashscore | 7.17 | 28.28 |
| Average B-factor | 18.20 | 112.54 |
| Macromolecules | 16.97 | 125.25 |
| Ligands | 9.66 | 110.32 |
| Solvent | 26.67 | 101.112 |

Statistics for the highest resolution shell are shown in parentheses.

concept showed that an XIAP-UBE2M chimera successfully NEDDylated XIAP substrates in cells (Zhuang et al, 2013). We assessed the ability of XIAP-RNF38 to form a complex with mature SMAC[56-C] by gel filtration chromatography (Fig 3B). In the elution profile of XIAP-RNF38/SMAC, species larger in size than SMAC[56-C] or XIAP-RNF38 were detected, supporting that XIAP-RNF38 chimeras bind SMAC. We compared ubiquitination of

mature SMAC by XIAP-RNF38 LP[Arg] and LP[Ala] (Fig 3C). XIAP-RNF38 LP[Ala] was deficient in attaching Ub to SMAC, with SMAC-Ub$_1$ only being detectable at the last time point measured (Fig 3C). In contrast, the LP[Arg] chimera produced SMAC-Ub$_1$ in less than 1 min and attached up to five Ubs to SMAC by the last time point. Among a panel of XIAP-RNF38 LP variants, LP[Arg] resulted in the most ubiquitinated SMAC products, whereas LP[Tyr], LP[Glu], and

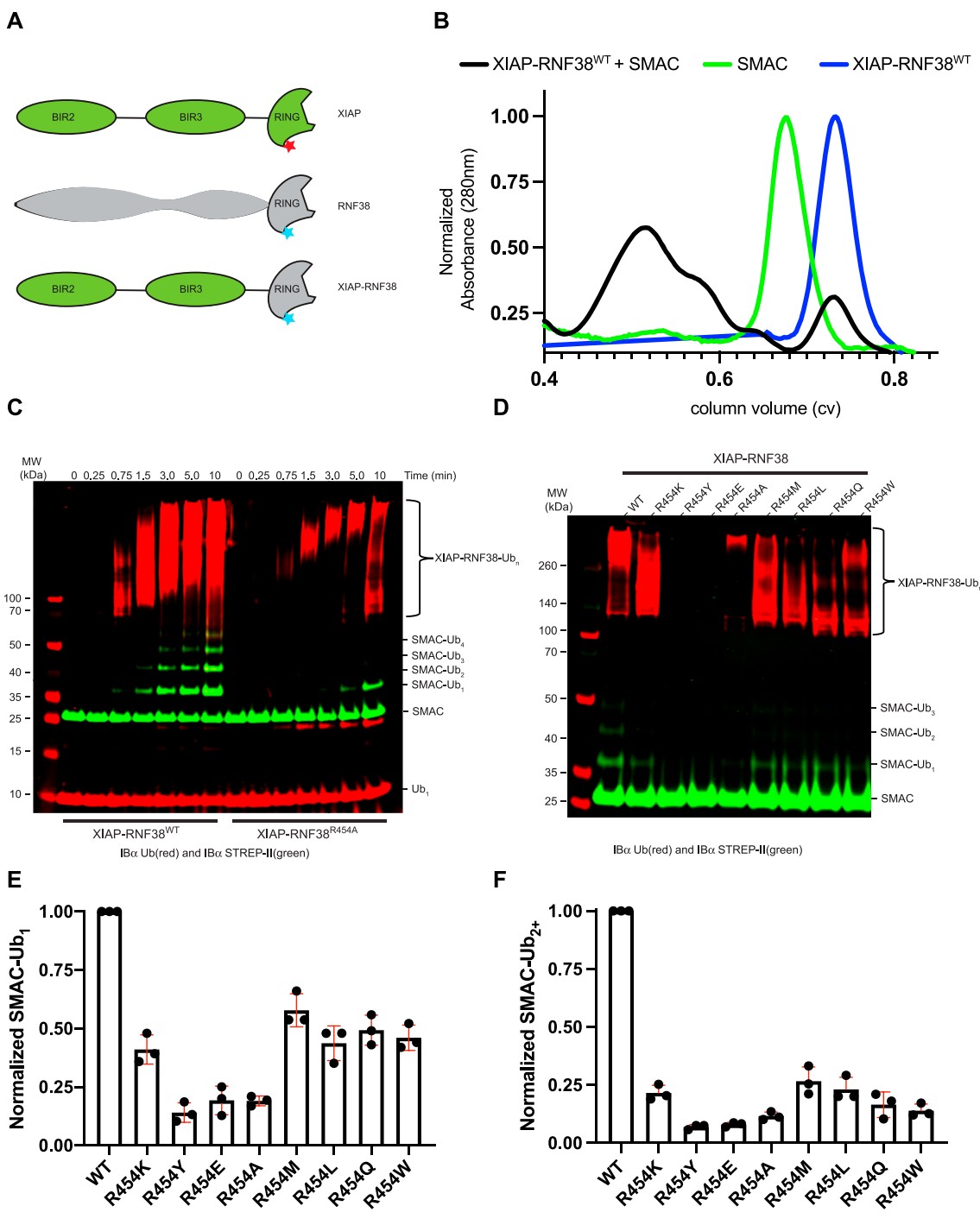

**Figure 3. Effect of LP on substrate ubiquitination in the XIAP-RNF38 chimera assay.**
**(A)** Schematic of RNF38 (gray), XIAP (green), and XIAP-RNF38 chimera with BIR2-BIR3 from XIAP (green), RNF38[RING] (gray), and the LP residue indicated with a star. **(B)** Gel filtration elution profiles of XIAP-RNF38[WT] (blue) and SMAC[56-C] (green) compared against E3 (XIAP-RNF38[WT])/SMAC complexes (black). **(C)** Western blot showing anti-StrepTag II (green) and anti-Ub (red) for XIAP-RNF38[WT] and XIAP-RNF38[R454A] ubiquitination of SMAC[56-C, Strep-II]. **(D)** Ubiquitination of SMAC[56-C, Strep-II] is detected by anti-StrepTag II (green) and anti-Ub (red) for indicated LP variants of XIAP-RNF38 at a single time point. **(E)** Amount of monoubiquitinated product, SMAC–Ub$_1$, is normalized to LP[Arg] for the indicated XIAP-RNF38 LP variants. **(F)** Signals corresponding to SMAC with 2-5 Ubs (SMAC–Ub$_{2-5}$) are normalized to LP[Arg] for each XIAP-RNF38 LP variant. Data for (E, F) are presented as individual data points and mean value ± SD from three independent experiments (n = 3).

LP[Ala] (R454Y, R454E, and R454A, respectively) produced the least (Figs 3D–F and S8A). Detection with a primary antibody for Ub (anti-Ub) revealed that some XIAP-RNF38 LP variants were also deficient in autoubiquitination (Fig 3C and D). This result is consistent with our previous assay that monitored depletion of mono-Ub through autoubiquitination (Fig 1E).

## Introducing a linchpin Arg enhances E3 ligase activity of XIAP

As revealed through multiple sequence alignment (Fig S1A), XIAP (also known as BIRC4) is a RING E3 ligase with $LP^{Tyr}$ rather than a cationic residue. To investigate the importance of the LP residue, we compared the E3 ligase activity of XIAP $LP^{Arg}$ (Y485R) and $LP^{Glu}$ (Y485E) variants for the native substrate, SMAC. $XIAP^{WT}$ was able to ubiquitinate SMAC better than $XIAP^{Y485E}$; however, XIA-$P^{Y485R}$ catalyzed ubiquitination of SMAC faster than either (Fig 4A). Analysis at a single time point revealed that $XIAP^{Y485R}$ produced ~33% more monoubiquitinated SMAC and extended the Ub chain more rapidly than $XIAP^{WT}$ or $XIAP^{Y485E}$ (Fig 4B–D). The activity of $XIAP^{Y485E}$ is only marginally reduced compared with $XIAP^{WT}$, which led us to investigate how XIAP LP variants affected $Ub^D$. We used $^1H,^{15}N$-HSQC to monitor $Ub^D$ in $UBE2D2^{S22R,C85K}$–Ub($^{15}N$) after the addition of XIAP RING LP variants. Binding to $Ub^D$ was clearly observed for both $XIAP^{WT}$ and $XIAP^{Y485R}$, whereas $XIAP^{Y485E}$ produced smaller CSPs and fewer signal attenuations (Figs 4E–G and S8B–D). Furthermore, $XIAP^{Y485R}$ produced more signal attenuations in $Ub^D$ and was the only variant to completely attenuate $Gln^{40}$ of $Ub^D$ (Fig 4H). The predicted structure of $XIAP^{Y485R}$ with $UBE2D2^{S22R,C85K}$–Ub suggested $Ub^D$ in the canonical primed conformation (Fig 4I). Although we have applied signal attenuations in a qualitative fashion, the clear spectral differences for both $RNF38^{WT}$ and $XIAP^{Y485R}$ support more residues of $Ub^D$ in the slow exchange regime. Attenuation of the $Gln^{40}$ cross-peak is indicative of the formation of the closed E2~Ub conformation, as $LP^{Arg}$ forms a hydrogen bond with $Gln^{40}$ of $Ub^D$ in the structures of RING E3/E2~Ub complexes (Fig 1B). Together, these results show that introduction of $LP^{Arg}$ facilitates the catalytically competent E2~Ub conformation and thereby enhances XIAP E3 activity.

## Effect of LP-activated XIAP in cells

Next, we wanted to investigate whether the gain in E3 activity of XIAP $LP^{Arg}$ as observed in our in vitro findings could be confirmed in a cellular environment. For this, $XIAP^{WT}$ or LP variant $XIAP^{Y485R}$ was overexpressed in HEK293 cells and the ubiquitination levels of a known XIAP substrate, PTEN (Van Themsche et al, 2009), were compared in their presence. Western blot analysis showed only a very modest level of PTEN ubiquitination in the presence of $XIAP^{WT}$, suggesting it to be a weak E3 ligase for PTEN. However, the expression of $XIAP^{Y485R}$ increased PTEN ubiquitination (Fig 5A), confirming that $XIAP^{Y485R}$ improved E3 activity for another substrate beyond SMAC. A cycloheximide chase experiment further corroborated this finding wherein we found that U2OS cells overexpressing $XIAP^{Y485R}$ showed a much-reduced PTEN half-life (~6 h) compared with cells overexpressing $XIAP^{WT}$ (~9 h) (Fig 5B). PTEN is a known negative regulator of the PI3K/AKT pathway (Ahmed et al, 2012), so we next checked whether the enhanced ubiquitination and degradation of PTEN in the presence of XIA-$P^{Y485R}$ lead to higher AKT activation. Indeed, as compared to $XIAP^{WT}$, the presence of $XIAP^{Y485R}$ resulted in higher levels of phospho-AKT $Ser^{437}$ (pAKT), concomitant with a reduced level of PTEN (Fig 5C). Furthermore, treatment with the proteasomal inhibitor MG132 rescued PTEN from $XIAP^{Y485R}$-dependent enhanced proteasomal degradation that in turn led to reduced levels of pAKT (Fig 5C). Altogether, these results demonstrate that incorporation of $LP^{Arg}$ in XIAP improves its E3 ligase activity for the bona fide substrate PTEN across different human cell types.

# Discussion

Our study demonstrates the important role of RING E3's LP residue in coordinating $Ub^D$ within the E2~Ub catalytic intermediate and introduces approaches to characterize RING-mediated E2~Ub binding and Ub transfer. By systematically investigating LP in the context of E2~Ub binding, E3 ligase activity, and the structural outcomes of LP substitutions, we have elucidated distinct roles of the LP residue in RING E3s. Initially, we hypothesized that the binding affinity of RING E3 to E2~Ub would strongly correlate with the activity of $Ub^D$ transfer. However, we found that LP variants with comparable affinities for E2~Ub exhibited as much as sixfold differences in activity. The representative structure of $RNF38^{RING}$-R454Y/$UBE2D2^{S22R,C85K}$–Ub along with our SPR and NMR results suggests that LP variants can still bind E2~Ub, but exhibit varying abilities to prime E2~Ub in the closed conformation. Our results demonstrated that the activity of $Ub^D$ transfer correlates with the E2~Ub priming ability of the amino acid at the LP position. These findings reinforce earlier studies indicating that the LP residue modulates the equilibria of E2~Ub conformations and Ub transfer activity. The E2~$^{15}N$-$Ub^D$ NMR assay coupled with E2~Ub binding analysis would be a useful approach to distinguish between unproductive binding and priming of $Ub^D$. Future application of $^{13}C$ NMR methods would provide more insights into the dynamics of these complexes and possibly overcome the loss of information from the slow exchange regime we observed using only $^{15}N$ labeling (Kleckner & Foster, 2011; Alderson & Kay, 2021).

Activation of XIAP through the single Y485R substitution suggests that alteration of the identity of LP residue could modulate the E3 activity. Substrate ubiquitination was enhanced with XIA-$P^{Y485R}$ in vitro and in cells. However, this is potentially a double-edged sword as excess autoubiquitination from an unnatural LP substitution could potentially lead to premature degradation of the E3. Analysis of all natural-occurring amino acids at the LP position of RNF38 showed that Arg is the most preferred residue when paired with the promiscuous E2 UBE2D2. Introduction of the XIAP-RNF38 chimera system provides an assay to directly compare the activity of isolated RING domains by eliminating the contribution from divergent substrate binding domains. Notably, the importance of the LP residue is equally valid in the related U-box E3s and other members of the RING E3 family. This is exemplified by a single mutation altering the activity of RBX1 RING, a component of the multisubunit cullin-RING E3 ligase (CRL) family (Paluda et al, 2022). The RBX1 LP N98R mutation stimulated activity with UBE2D2, whereas $RBX1^{N98A}$ inhibited this activity (Middleton et al, 2020). Other structural studies have guided the choice of LP mutation to validate its role (Furukawa et al, 2000; Metzger et al, 2014; Buetow et al, 2015).

Another important consideration is the cooperativity of the E2/E3 pair (Komander & Rape, 2012; Plechanovová et al, 2012; Middleton

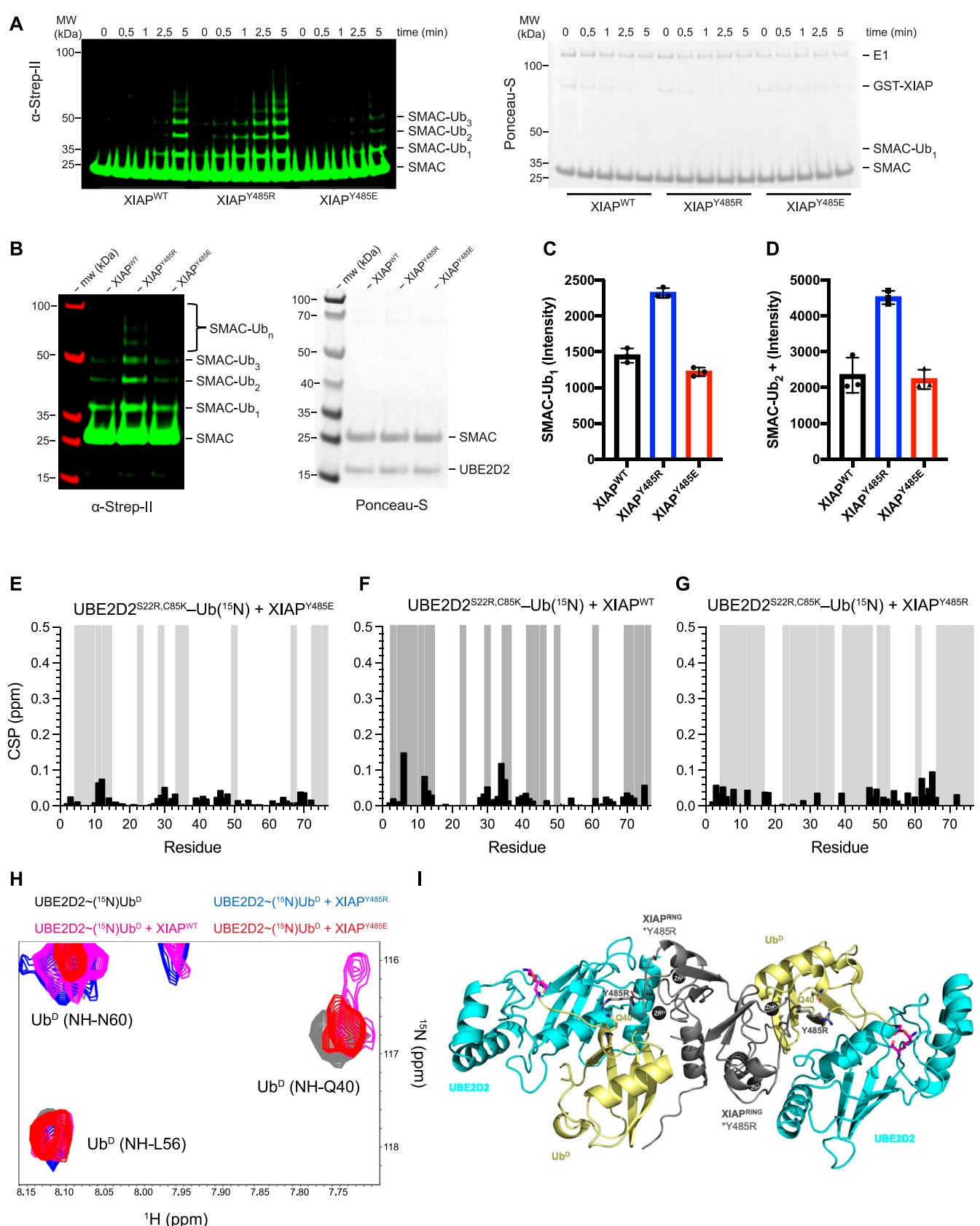

et al, 2017; George et al, 2018). For example, c-Cbl crystallized with UBE2L3 (UbcH7), yet this E2/E3 pair is deficient in ubiquitination compared to c-Cbl pairing with the UBE2D family E2s (Dou et al, 2012a; Branigan et al, 2015; Nayak & Sivaraman, 2018). With limited prior knowledge, it is challenging to determine which E2/E3 pairs will be productive or nonproductive (Marblestone et al, 2013). Nevertheless, the numerous structures of RING E3s in complex with different E2~Ub conjugates conserve $Ub^D$ in the primed conformation, highlighting the central role of the LP residue. It remains open if the LP residue contributes to selection of UBE2D family and related E2s. With ~40 E2s in humans, there are always exceptions such as UBE2T, which does not require $Ub^D$ in a closed conformation for catalysis (Chagule et al, 2020). Clearly, RING domains have evolved to recognize many binding partners aside from E2~Ub (Kiss et al, 2019; Nakasone et al, 2022). Interestingly, somatic mutations in RING E3 LP residues have been detected in human cancers, suggesting a link between disease and E3 ligase activity (George et al, 2018; Duan & Pagano, 2021). Post-translational modifications in proximity of the LP residue also merit exploration as demonstrated by acetyl-lysine in UHRF1 (Nihira et al, 2017; Lacoursiere et al, 2022; Wang et al, 2024). A similar outcome could be possible with binding partners such as XAF1, which binds the LP residue, $Tyr^{485}$, in XIAP and reduces E3 activity (Tse et al, 2012). Structural insights into CBL and MDM2 show how single phosphosites near the RING domain can also contribute to stabilizing $Ub^D$ (Dou et al, 2013; Magnussen et al, 2020). Furthermore, phage display libraries have yielded proteins with specificity for a single RING domain that can be guided to either inhibit or stimulate E3 activity (Gabrielsen et al, 2017; Watson et al, 2019).

Although it is not clear whether RING domains have druggable sites for small molecules, the LP residue and E2~$Ub^D$ interface provide a target and mechanism for RING E3 regulation. Such mechanistic understanding is applicable to the emerging field of custom E3 ligases and would ensure optimal ubiquitination of both natural and unnatural targets (Zhuang et al, 2013; Ibrahim et al, 2020; Zeng et al, 2021). Therefore, designing such potent E3 ligases would help overcome some of the major challenges facing the targeted protein degradation (TPD) field (Alabi & Crews, 2021; Li & Crews, 2022; Webb et al, 2022). In addition, our experimental framework and general concepts for optimal RING domains could guide the emerging field of generative protein design for de novo E3 ligases.

It remains an open question as to why some RING E3s have suboptimal residues in the LP position. XIAP and cIAP1 bind the same substrates; however, despite its reduced RING E3 activity, XIAP potently inhibits caspases through substrate binding domains in contrast to cIAP1 (Takahashi et al, 1998; Mufti et al, 2007; Choi et al, 2009; Gill et al, 2009; Lalaoui & Vaux, 2018). Another consideration is RING heterodimers that have evolved one inactive scaffold and one active E3 as found in MDM2/MDM4 and

RNF2/Bmi1 (Li et al, 2006; Nomura et al, 2017). Whether such systems serve to provide an internal mechanism to regulate ubiquitination, substrate recruitment, E2~Ub recruitment, or other functions merits exploration. A recent study has shown that although the role of the LP residue is conserved in yeast ERAD E3s Hrd1 and Doa10, the identity of the LP residue influences the E2 recruited and subsequently the kinetics of ubiquitination (Lips et al, 2020), highlighting the necessity of investigating RING E3 activity on a case-by-case basis. Our study primarily investigated the role of the LP residue with a single E2, UBE2D2. Whether the nature of the LP residue has any influence on other E2s requires further investigation.

In summary, we have extensively assessed how the LP residue affects ubiquitination and found that this single residue is crucial for coordinating $Ub^D$ and stabilizing an optimal conformation of $Ub^D$ for transfer. The LP residue plays a critical role in regulating the overall activity of RING E3 ligases, and mutations of this residue can increase or decrease activity in vitro and in cells. These mechanistic insights provide new avenues for designing molecules to control RING-mediated ubiquitination and how RING E3s function.

## Materials and Methods

### DNA constructs and design

RING E3s $XIAP^{145-C}$ and $RNF38^{389-C}$ and Cys-Ub for fluorescent labeling were expressed as GST-TEV fusions using pGEX4T-1 (Buetow et al, 2015). Point mutations for LP residues were introduced by site-directed mutagenesis (SDM) using primer pairs to introduce each point mutation. Human E1 (UBA1) for Ub was expressed according to Nakasone et al (2022), and E2s were expressed following the protocol described previously (Dou et al, 2012b; Buetow et al, 2015). Mature human $SMAC^{56-C}$ with a C-terminal StrepTag II was expressed as a SUMO fusion in pRSF-DUET1 to ensure the N terminus started from $Ala^{56}$.

### Protein expression and purification

Proteins were expressed in *Escherichia coli* BL21 (DE3) Rosetta 2 pLysS (Novagen). Cultures were grown in Luria–Bertani (LB) medium at 37°C to an $OD_{600}$ of 0.6–0.8 and induced with 0.25 mM IPTG at 20°C for 12–16 h. $ZnSO_4$ was added to a final concentration of 200 $\mu$M before induction of RING E3s. Cells were harvested by centrifugation, resuspended in IMAC buffer (25 mM Tris–HCl, 40 mM imidazole, 500 mM NaCl, pH 7.5) for His or GST buffer (25 mM Tris–HCl, 500 mM NaCl, 5 mM DTT, pH 7.5), and lysed

**Figure 4. Activating XIAP through LP substitution in vitro.**
**(A)** Ubiquitination of $SMAC^{56-C,Strep-II}$ by $XIAP^{WT}$, $XIAP^{Y485R}$, and $XIAP^{Y485E}$ is detected by anti-StrepTag II (green) over the indicated time course. **(B)** Single time point of $SMAC^{56-C,Strep-II}$ ubiquitination with indicated XIAP LP variants. **(C, D)** Intensity of (C) monoubiquitinated $SMAC^{56-C}$–$Ub_1$ and (D) $SMAC$–$Ub_{2-5}$ is plotted for each XIAP LP variant from three independent experiments (n = 3) with individual data points and mean value ± SD. **(E, F, G)** Residue-specific chemical shift perturbations (black) and signal attenuations (gray) of $UBE2D2^{S22R,C85K}$–$Ub^D$ ($^{15}N$) with a 1:1 ratio of (E) $XIAP^{Y485E}$, (F) $XIAP^{WT}$, and (G) $XIAP^{Y485R}$. **(H)** $^1H$-$^{15}N$ HSQC overlay showing the $Gln^{40}$ region of $Ub^D$ in $UBE2D2^{S22R,C85K}$–$Ub^D$ ($^{15}N$) alone (black) and a 1:1 M ratio of $XIAP^{WT}$ (gray), $XIAP^{Y485R}$ (blue), and $XIAP^{Y485E}$ (red). **(I)** AlphaFold3 (Abramson et al, 2024) model of homodimeric $XIAP^{Y485R}$ RING domain (gray) with $UBE2D2^{S22R,C85K}$–Ub; UBE2D2 (cyan), $Ub^D$ (yellow), $Gly^{76}$ of $Ub^D$ and $Lys^{85}$ of $UBE2D2^{S22R,C85K}$ (magenta sticks), $Zn^{2+}$ ions (dark gray spheres), and $Gln^{40}$ of $Ub^D$ and $Arg^{485}$ of XIAP as sticks.

**Figure 5. XIAP is activated by LP substitution in cells.**
**(A)** Immunoblots of PTEN ubiquitination from lysates of HEK293 cells transfected with plasmids expressing HA-XIAP variants or empty vector (EV) along with GFP-PTEN and His-Ub and treated with MG132. The Ni-NTA pull-down products and cell lysates were analyzed by immunoblotting using anti-GFP, anti-HA, or anti-actin antibodies as indicated. Actin was used as the loading control. **(B)** Immunoblots showing the stability of PTEN from lysates of U2OS cells expressing HA-XIAP variants treated with cycloheximide for indicated times. The cell lysates were analyzed by immunoblotting using anti-PTEN, anti-HA, or anti-actin antibodies as indicated. Actin was used as the loading control. **(C)** Immunoblots showing the effects of XIAP variants on PTEN and activated AKT. U2OS cells were transfected with plasmids expressing HA-XIAP variants and left untreated or treated with MG132. Lysates were analyzed by immunoblotting using anti-pAKT (S473), anti-AKT, anti-PTEN, anti-HA, or anti-actin antibodies as indicated. Actin was used as the loading control. The experiments were performed in three biological replicates.

at 12,000 psi using a microfluidizer. The lysates were cleared by centrifugation at 36,000$g$, filtered through a 0.45-$\mu$m syringe unit, and applied to a 5-ml His-TRAP (GE Lifesciences) or GST column. Protocols for generating UBD2D2 and related mutants (Dou et al, 2012b), C-terminal hexahistidine-tagged human Uba1-6xHis (Nakasone et al, 2022), and ubiquitin (Varadan et al, 2002; Pickart & Raasi, 2005; Nakasone et al, 2013, 2022) have been published previously.

His-GST fusions with TEV sites were cleaved at 4°C overnight with 1:100 (w/w) TEV protease. RING domains remained stable after the first affinity capture step, negating the need for additional zinc. The 6xHis or His-SUMO proteins were purified using a 5 ml His-TRAP crude (GE Lifesciences). Further purification was performed by size-exclusion chromatography on a Superdex 75 column (GE Healthcare) pre-equilibrated in 25 mM Tris–HCl, pH 7.6, 150 mM NaCl, 1 mM DTT. Protein concentrations were determined either by measurement of the absorbance at 280 nm based on molar extinction coefficients calculated from the relevant sequences using Expasy's ProtParam (https://web.expasy.org/protparam/) or by the Bradford assay.

### Monoubiquitin depletion assay

GST-RNF38 LP variants were quantified by the Bradford assay, and 1 $\mu$g of each variant was visualized by SDS–PAGE followed by staining with Quick Coomassie (Generon). Concentrations of LP variants were normalized to GST-RNF38 WT. Autoubiquitination assays were carried out in 50 mM NaCl, 50 mM Tris–HCl, pH 7.6, 5 mM ATP, and 5 mM MgCl$_2$ with 0.5 $\mu$M UBA1, 50 $\mu$M Ub, 5 $\mu$M UBE2D2, and 1 $\mu$M GST-RNF38 LP variant at 23°C. All components except GST-RNF38 LP variant were mixed and incubated for 2 min at 23°C. The time course was initiated upon the addition of the GST-RNF38 LP variant and quenched at the indicated time points with the addition of SDS–PAGE loading dye. For the time course control reactions, no Mg$^{2+}$-ATP was included. Samples were treated with 50 mM DTT for 5 min before separation by SDS–PAGE, imaging on a LI-COR Odyssey CLx, and staining with Quick Coomassie (Generon). Ub depletion was quantified in three independent experiments at a single time point for each LP variant (WT, 2 min; R454K, R454M, R454W, 5 min; R454Q, 10 min; R454A, R454Y, R454E, 20 min). Depletion was measured relative to the amount of mono-Ub at the same time point in a reaction containing no E3 ligase.

### XIAP-RNF38 chimera and XIAP LP variant assay with mature SMAC

Ubiquitination of SMAC by XIAP-RNF38 was carried out in assay buffer (25 mM Tris–HCl, 150 mM NaCl, pH 8.0) with 15 mM ATP, 15 mM MgCl$_2$, 1 $\mu$M UBA1, 200 $\mu$M Ub, and 10 $\mu$M UBE2D2 at 37°C. The concentration of each XIAP-RNF38 variant was standardized and held at 1 $\mu$M for assays. Reactions were stopped at the indicated time points with LDS loading buffer and resolved by reducing SDS–PAGE. Gels were transferred for 10 min with constant 1.3 Amps to a nitrocellulose membrane (Bio-Rad) using the Trans-Blot Turbo

system (Bio-Rad). Total protein was visualized with Ponceau-S staining and destained, and the membrane was blocked in 5% (wt/vol) BSA in TBST (20 mM Tris–HCl, 150 mM NaCl, 0.1% [wt/vol] Tween-20) for 45 min at ambient temperature. Primary antibodies were incubated at 4°C for 14 h using a 1:4,000 dilution of mouse anti-Ub (PD41; Santa Cruz Biotechnology) and a 1:2,000 dilution of mouse StrepTag II monoclonal (71590-3; Merck Millipore) primary antibody in 2.5% (wt/vol) BSA in TBST. The membranes were washed in TBST three times for 5 min before incubation with secondary antibodies, IRDye 800CW goat anti-mouse IgG and IRDye 680RD goat anti-rabbit IgG (LI-COR Biosciences), for 1 h. Two washes with TBST and then TBS were carried out before imaging on a LI-COR Odyssey CLx.

### Conjugation of near-infrared fluorescence–labeled Ub

The GGSC-Ub was reduced with 20 mM TCEP and exchanged to 25 mM Hepes, 150 mM NaCl, pH 7.5, using a 0.5-ml Zeba Spin desalting column (cat. no. 89882; Thermo Fisher Scientific). IRDye 680RD Maleimide (LI-COR Biosciences) was resuspended in DMSO and added at a threefold molar excess to GGSC-Ub and allowed to incubate at room temperature for 90 min. Labeling was quenched with the addition of 3 mM BME, and excess dye was removed and buffer-exchanged to 25 mM Tris–HCl, 150 mM NaCl, pH 8.0, using a 0.5-ml Zeba Spin desalting column. The final stock 200 $\mu$M of IR-labeled GGSC-Ub (Ub*) was mixed with unlabeled Ub$^{WT}$ in 1:80 M ratio to obtain a working stock.

### Crystallization of RNF38 and UBE2D2 RNF38 complexes

All crystals were obtained using vapor diffusion at 18°C and commercially available screens. The high-resolution structure of RNF38 was obtained in condition 16 in the MORPHEUS screen (Molecular Dimensions; 0.09 M halogens, 0.1 M Buffer System 1, pH 6.5, 37.5% vol/vol Precipitant Mix 4). RNF38 mutants were mixed in a 1:1 M ratio with UBE2D2-Ub, and crystallized in condition 93 in the JCSG+ screen (Molecular Dimensions; 0.2 M sodium acetate trihydrate, 0.1 M imidazole, pH 8.0, 10% PEG 8000).

### X-ray data collection, structure determination, and refinement

Data were collected at Diamond Light Source and processed using DIALS as part of the XIA2 pipeline. Structures were determined by molecular replacement using PHASER and the structure of RNF38 WT alone or in complex with UBE2D2$^{S22R}$–Ub (PDB 4V3K). Refinements were done in BUSTER (Global Phasing) or PHENIX (Liebschner et al, 2019), and visual inspection and manipulation were done in COOT (Emsley et al, 2010).

### Solution NMR experiments

Solution NMR data were acquired on a Bruker Avance III HD 600-MHz spectrometer with a cryogenic TCI probe. To observe Ub$^D$ in the E2 conjugates, uniformly labeled Ub($^{15}$N) was loaded on UBE2D2$^{S22R,C85K}$ to generate UBE2D2$^{S22R,C85K}$–Ub($^{15}$N) following the protocol described previously (Buetow et al, 2015). To ensure

binding could be observed, the concentration of UBE2D2$^{S22R,C85K}$–Ub($^{15}$N) never fell below 50 $\mu$M—a regime well above the $K_d$ for reported E2~Ub/RING interactions (Gabrielsen et al, 2017). In the final step of purification, UBE2D2$^{S22R,C85K}$–Ub($^{15}$N) was run on a Superdex S75 16/600 column in 20 mM sodium phosphate (pH 7.0), 100 mM NaCl. The measurements were carried out in 3-mm NMR tubes starting with UBE2D2$^{S22R,C85K}$–Ub($^{15}$N) at 320 $\mu$M for RNF38 and at 100 $\mu$M for XIAP. To maintain a lock signal, D$_2$O was added to 7.5% (vol/vol) for each sample. The $^{15}$N-$^1$H-fast-HSQC spectra (Mori et al, 1995) were recorded with 128 real points in the $^{15}$N dimension and processed using 256 points with linear prediction in Bruker TopSpin version 3.5 patch level 7. An intermediate titration point of 0.5:1, XIAP$^{RING}$:UBE2D2$^{S22R,C85K}$–Ub($^{15}$N), was recorded but proved unhelpful for analysis because of binding of XIAP$^{Y485R}$ shifting signals in UBE2D2$^{S22R,C85K}$–Ub($^{15}$N) to the slow exchange regime. In addition, the end point at 1:1 was also recorded with 256 scans for XIAP$^{WT}$ and XIAP$^{Y485R}$ in an attempt to recover signals from Q40 in donor Ub. With XIAP$^{WT}$, the signal for Q40 was observed in two states, whereas we could not recover any signal for Q40 with XIAP$^{Y485R}$ at 1:1. All spectra were analyzed using CARA and NMRFAM-SPARKY (Lee et al, 2015). Signals that fell to the noise in $^1$H,$^{15}$N-HSQC spectra were classified as attenuated and were assessed at several threshold and contour levels.

The CSP was calculated according to

$$CSP = \left[ (\delta_{HA} - \delta_{HB})^2 + ((\delta_{NA} - \delta_{NB})/5)^2 \right]^{1/2}.$$

### SPR binding and analysis

SPR binding experiments were performed at 25°C on a Biacore T200 instrument using a CM-5 chip (GE Healthcare) with coupled anti-GST nanobody as described previously (Ahmed et al, 2020). GST-tagged RNF38 LP variants were captured, and a serial dilution of UBE2D2$^{S22R}$ or UBE2D2$^{S22R,C85K}$–Ub in running buffer containing 25 mM Tris–HCl, pH 7.6, 150 mM NaCl, 1 mM DTT, and 0.005% (vol/vol) Tween-20 was used as an analyte. Two technical replicates were performed, data were analyzed and fit to a one-site binding model with BIAevaluation (GE Healthcare), and the data were plotted in Prism (GraphPad version 8).

### Mammalian cell culture and transfection

The mammalian cell lines HEK293 and U2OS were purchased from the ATCC and cultured in a monolayer in 5% CO2 at 37°C in DMEM supplemented with 10% FBS, 20 mM L-glutamine, 100 units/ml penicillin, 0.1 mg/ml streptomycin, and 6 mg/liter gentamycin (Thermo Fisher Scientific). The cell lines were authenticated in-house by short tandem repeat profiling using GenePrint 10 System (Promega) every 2 yr. Regular in-house *Mycoplasma* testing was also carried out on the cells. The cells were transfected with the indicated plasmids using JetPEI DNA transfection reagent (Polyplus transfection) following the manufacturer's protocol. The cells were harvested 48 h post-transfection unless otherwise mentioned.

## Antibodies and chemicals

The primary antibodies used in this study include mouse anti-GFP (cat. no. sc-81045, 1:1,000 for Western blot; Santa Cruz Biotechnology), rabbit anti-HA (cat. no. 3724, 1:1,000 for Western blot; Cell Signaling), mouse anti-actin (cat. no. sc-47778, 1:1,000 for Western blot; Santa Cruz Biotechnology), rabbit anti-PTEN (cat. no. ab170941, 1:1,000 for Western blot; Abcam), mouse anti-pan AKT (cat. no. 2920, 1:1,000 for Western blot; Cell Signaling), rabbit anti-pAKT Ser473 (cat. no. 4060, 1:1,000 for Western blot; Cell Signaling). The following secondary antibodies were used: goat anti-rabbit IRDye 680LT (cat. no. 925-68021, 1:20,000 for Western blot; LI-COR Biosciences), goat anti-mouse IRDye 680LT (cat. no. 925–68020, 1:20,000 for Western blot; LI-COR Biosciences), and goat anti-mouse IRDye 800CW (cat. no. 925–32210, 1:15,000 for Western blot; LI-COR Biosciences). Cycloheximide (Sigma-Aldrich) was dissolved in water to make a stock concentration of 25 mg/ml, whereas MG132 (Merck Millipore) was dissolved in DMSO to make a stock concentration of 50 mM. Before harvesting, the cells were treated with 50 $\mu$M MG132 for 4 h in the cell-based ubiquitination profiling assay and where indicated in the cycloheximide chase assay.

## Cell-based ubiquitination profiling

The plasmids were transfected into HEK293 cells to overexpress the respective proteins as shown in the figures. The ubiquitination assay was performed in the presence of 8 M urea. Cell lysis was carried out in ubiquitination buffer A (UBA) (8 M urea, 0.3 M NaCl, 50 mM phosphate, pH 8.0, 100 $\mu$g ml$^{-1}$ N-ethylmaleimide). The samples were then sonicated and centrifuged, and the resulting lysates were mixed with Dynabeads His-tag matrices (Invitrogen) and incubated at 4°C overnight on a rotatory shaker. The next day, samples were processed following the previous protocol (Ahmed et al, 2021). SDS–PAGE was used to separate the Ni$^{2+}$–pull-down products, and immunoblotting was performed to detect the ubiquitinated adducts using specific antibodies.

## Cycloheximide chase assay

The cycloheximide chase experiment was performed following the protocol as described before (Ahmed et al, 2015). Briefly, U2OS cells were transfected with the plasmids as indicated in Fig 5B. After 48 h, the cells were treated with 50 $\mu$g/ml cycloheximide in fresh medium. The cells were then harvested at indicated time points as shown in Fig 5B, and whole-cell lysates were prepared. The stability of the target substrate PTEN was checked in the presence of the E3 ligase XIAP or its variant XIAP–Y485R through immunoblotting.

## Cell lysis and immunoblotting

The cells were lysed in cell lysis buffer containing 50 mM Tris–HCl, pH 7.2, 150 mM NaCl, 10% (vol/vol) glycerol, 1% (vol/vol) IGEPAL CA-630, protease inhibitor cocktail, 2.5 mM PMSF, 1 mM DTT, and 1 mM EDTA as described before (Ahmed et al, 2015). Fifty micrograms of protein per lane was loaded for immunoblotting of whole-cell lysates. SDS–PAGE was performed using NuPAGE 4–12% Bis-Tris gels (Thermo Fisher Scientific) to separate the protein samples, and immunoblotting was performed following the established protocol (Magnussen et al, 2020).

## Quantification and statistical analysis

In-gel fluorescence was acquired using an Odyssey CLx near-infrared imaging instrument with ImageStudio v5 and analyzed with ImageStudio v5 to quantify Ub depletion or the Fiji distribution of ImageJ[32] for SMAC ubiquitination. Data were plotted using Prism 9 (GraphPad). The following equation was used to calculate the fraction of mono-Ub depleted per minute:

$$(1 - I_{\text{LP variant}} / I_{\text{No E3}}) / t,$$

where $I$ is the fluorescent intensity of the mono-Ub band at the selected time point.

# Data Availability

Coordinates and structure factors are deposited with PDB 9Q8Y and 9Q88. Associated raw data are deposited on Zenodo (https://doi.org/10.5281/zenodo.15178736), including supporting data for response to reviewers.

# Supplementary Information

# Acknowledgements

We thank DLS for access to beamline I04 and I04-1 (proposal mx16258) that contributed to the results presented here; W Clark and A Keith for in-house DNA sequencing, and Catherine Winchester for critical reading of the article (CRUK Scotland Institute). We thank Kevin Haubrich for discussion on NMR data. This work was supported by the Cancer Research UK core funding to the CRUK Scotland Institute (A17196 and A31287), CRUK core program funding to DT Huang (A29256), and European Research Council (ERC) under the European Union's Horizon 2020 research and innovation program (grant agreement no. 647849) to DT Huang.

## Author Contributions

MA Nakasone: conceptualization, resources, data curation, software, formal analysis, supervision, funding acquisition, validation, investigation, visualization, methodology, project administration, and writing—original draft, review, and editing.
L Buetow: conceptualization, resources, data curation, software, formal analysis, supervision, validation, investigation, visualization, methodology, project administration, and writing—original draft, review, and editing.
M Gabrielsen: conceptualization, resources, data curation, software, formal analysis, validation, investigation, visualization,

methodology, project administration, and writing—original draft, review, and editing.

SF Ahmed: resources, formal analysis, investigation, methodology, project administration, and writing—original draft, review, and editing.

KA Majorek: resources, formal analysis, validation, investigation, and writing—original draft, review, and editing.

GJ Sibbet: resources, data curation, validation, investigation, visualization, and methodology.

BO Smith: data curation, supervision, validation, investigation, visualization, methodology, and writing—original draft, review, and editing.

DT Huang: conceptualization, resources, data curation, formal analysis, supervision, investigation, project administration, and writing—original draft, review, and editing.

## Conflict of Interest Statement

The authors declare that they have no conflict of interest.

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
