## [Reviewer comments · Life Science Alliance]

Life Science Alliance

Tuning ubiquitin transfer by RING E3 ubiquitin ligases through the linchpin residue

Mark Nakasone, Danny Huang, Lori Buetow, Mads Gabrielsen, Syed Ahmed, Karolina Majorek, Gary Sibbet, and Brian Smith
DOI: <https://doi.org/10.26508/lsa.202503394>

Corresponding author(s): Danny Huang, Cancer Research UK Beatson Institute; Danny Huang, Cancer Research UK Beatson Institute; and Mark Nakasone, Cancer Research UK Beatson Institute

Review Timeline:

Submission Date:	2025-05-20
Editorial Decision:	2025-06-20
Revision Received:	2025-07-06
Editorial Decision:	2025-07-08
Revision Received:	2025-07-10
Accepted:	2025-07-14

Scientific Editor: Tim Fessenden

Transaction Report:

June 20, 2025

RE: Life Science Alliance Manuscript #LSA-2025-03394-T

Dr. Mark A. Nakasone
Cancer Research UK Beatson Institute
Ubiquitin
Garscube Estate
Switchback Road
Glasgow, Scotland, UK G61 1BD
United Kingdom

Dear Dr. Nakasone,

Thank you for submitting your manuscript entitled "Tuning ubiquitin transfer by RING E3 ubiquitin ligases through the lynchpin residue" to Life Science Alliance. This manuscript was assessed by three expert reviewers, whose comments are below.

As you will see, reviewers were unanimous in their appreciation of the careful analysis of the lynchpin residue of this E3 ligase, with unexpectedly high ligase activity retained in some of the mutants. Reviewers 1 and 2 requested changes to the text to improve the discussion and rationale for some of the methods. Reviewer 3 was supportive but felt that some claims based on NMR analysis should be strengthened, for instance by expressing peak attenuation as a ratio of intensities. Given the overall strong support, we invite you to submit a revised manuscript that addresses all minor concerns, addresses the major concerns of Reviewer 3 in a manner of your choosing, and will not be subject to further peer review. We would be happy to publish your paper in Life Science Alliance pending these revisions as well as some changes necessary to meet our formatting guidelines.

- Please be sure that the authorship listing and order is correct.
- Please upload all figure files as individual ones, including the supplementary figure files; all figure legends should only appear in the main manuscript file.
- Please add a Running Title and a Summary Blurb/Alternate Abstract in our system.
- Please add ORCID ID for secondary corresponding author - they should have received instructions on how to do so.
- Please add a Category for your manuscript in our system.
- Please add the X and Bluesky handles of your host institute/organization as well as your own or/and one of the authors in our system.
- Please be sure that the authorship order is correct and matches between the system and the manuscript file.
- Please rename your sections according to our guidelines and be sure to include all of them as listed here: Title page, Summary blurb, Abstract, Introduction, Results, Discussion, Materials and Methods, Data Availability, Acknowledgements, Author contributions, Conflict of interest, References, Figure legends, including supplementary figure legends.
- Please add an Author Contributions section to the system as well.
- Please add your main, supplementary figure, and table legends to the main manuscript text after the references section.
- Please label the panels in Figure S3 accordingly.
- We encourage you to revise the figure legend for Figure S8 such that the figure panels are introduced in alphabetical order.
- Please add callouts for Figures 4I; S3A-B; S6A-C and S7A-D to your main manuscript text.

A. FINAL FILES:

B. MANUSCRIPT ORGANIZATION AND FORMATTING:

Sincerely,

Reviewer #1 (Comments to the Authors (Required)):

The manuscript describes the importance of the linchpin arginine (LP-Arg) residues on loop two on RING-type E3 ligases using the RNF38 RING E3 ligase protein. To perform these studies, the group mutate the LP-Arg to the 19-other amino acids and test their activity via autoubiquitination and perform NMR studies as well as co-crystallization with several mutants. The for various mutant more polyubiquitinated product, the more active the enzyme. Ultimately, the result of this extensive work confirms the importance of LP-Arg in Ub chain elongation by RING E3 ligases. To perform some of these studies, they solved the first structure of the RING domain of RNF38. In addition, they solved the structure of RING domain in complexed with the E2 enzyme covalently attached to the active site Ub. Lastly, they use the XIAP RING protein with weak activity and engineer a RING - protein with the binding domain of XIAP substrate and demonstrate that they can polyubiquitinate that protein. They furthermore introduced the LP0-Arg in XIAP which lacks an arginine and demonstrate substrate ubiquitination by XIAP-Arg.

As a whole, these is an extensive body of work that confirms the importance of the H-bond between a glutamine adjacent to the catalytic cysteine on the E2 and this LP-Arg on loop2. However, its work publishing but hopefully with minor important that should highlight the extensive work of the manuscript

I would have like to see greater discussion on the structure of the RING domain, as RING domains are not common, purification

conditions of wt and mutants, and how its structure may be different or similar to other RING and the location of the LP-Arg. I would have also liked greater discussion on the complexes unless this is repeated work of others and a better and confirm others works, which then takes away from the importance; thus the authors should highlight novel finding. If the authors chose not to highlight some of this exciting work, then it would reflect confirming work of several published work. In conclusion, I don't have a major criticism of this manuscript. Its well written, contains a lot of data and effort that I think needs described and highlight in the manuscript and not in supplemental material. As presented, its seems to target a very narrow and small subset of ubiquitination community.

Reviewer #2 (Comments to the Authors (Required)):

Interesting study that looks at the importance of and the chemical nature of the amino acid on the linchpin position in RING E3-ligases. It is to be expected that deviation from Arg to anything else (RNF38) will result in a drop of affinity and potentially also activity; strikingly this is not the case for all mutations; and a molecular explanation is giving based on NMR-experiments for one deviating mutant. In addition work on a XIAP-RNF38 chimera reveals that when replacing the native Tyr with an Arg an increase in activity is seen. Also activity in vivo is studied, to start casting a light on the cell biological importance of LPs.

Based on my evaluation the manuscript could be well suited for publication after some minor clarifications have been addressed. The study is a bit fragmented as studies on multiple ligases (RNF38 and XIAP-RNF38 chimera) is presented, also multiple cell lines are used for in vivo work.

-Can the authors give a rational why initially XIAP induced SMAC Ubiquitination is monitored and on a later stage XIAP induced PTEN ubiquitination?

-Can the authors give a rational for why initially HEK293s are used and in the chx experiment U2Os are used?

Reviewer #3 (Comments to the Authors (Required)):

Comments on Tuning ubiquitin transfer by RING E3 ubiquitin ligases through the linchpin residue by Nakasone et al

RING E3s represent a major class of E3s that rely on their catalytic RING domain to interact with the E2~Ub conjugate. A key feature of this domain is a conserved arginine residue, often referred to as the 'linchpin arginine,' which stabilizes the E2~Ub in a closed, catalytically competent conformation, thus promoting efficient ubiquitin transfer.

This manuscript investigates the active role of linchpin residue in mediating ubiquitin transfer and shows how different residues at this position modulate ubiquitination efficiency to varying degrees by differentially stabilizing the E2~Ub closed conformation. The authors suggest a combination of binding and NMR experiments to evaluate activation of E2 by RING-E3s.

The linchpin residue has been essential for understanding the activity of ubiquitin E2/E3 enzymes and has broad implications for protein regulation. The findings of the manuscript will be of interest to the community. I have a few concerns regarding the interpretation of the data.

Major Comments

1) Linchpin arginine establishes an intricate Hydrogen bond network with E2 and Ub to stabilize the closed conformation, where arginine acts as an H-donor. In the paper, substitutions like methionine, a non-H-bonding residue, support higher ubiquitination efficiency compared to serine or tyrosine, which are capable of H-bonding. This confuses the mechanistic insight into how LP substitution modulates activity.

The NMR experimental CSPs of E2~Ub(15N)/RNF38-R454M complex and E2~Ub(15N)/RNF38-R454S may shed some light.

2) I have a few concerns regarding the discussion of the NMR assay and data in the manuscript, which are listed below.

i) Despite the E2~Ub/WT-RNF38 adopting a single closed conformation, why do some peaks broaden while others do not? This point remains unclear and has not been addressed.

ii) It is better to present peak attenuation as a ratio of intensities (I/I_0) to better understand the data. This would clarify a few ambiguous texts in the document, such as "XIAP Y485R produced more signal attenuations in UbD and was the only variant to completely attenuate Gln40 of UbD."

iii) The number of attenuated signals has been repeatedly used in the text as a sensitive measure of activation and closed conformation. In contrast, it is a very poor measurement that provides no quantification. I strongly suggest removing such statements from the text. Or the authors need to provide a quantification of closed/open conformations.

Minor Comments

1. Some of the subheadings for "results" section can be reconsidered to convey the key findings of the experiment. For example, "Effect of LP variants on ubiquitination" can be rephrased to "LP variants differentially modulate ubiquitination efficiency".

Other subheadings that can be rephrased include:

- "Binding of RNF38 LP mutants to UBE2D2 and UBE2D2-Ub"
- "Structure of E2-Ub with RNF38 RING LP variants"

- "Effect of LP-activated XIAP in cells"

2. There is a discrepancy in the statement "The deficiency of the R454Y variant in the ubiquitination assay compared to the R454K, R454M, and R454W variants was unexpected as it exhibited the second-best binding affinity to UBE2D2S22R, C85K-Ub after RNF38RING-WT".

According to table 1 depicting affinities of E2 and E2-Ub conjugate, both R454W and R454Y have similar affinities for E2-Ub conjugate.

3. Supplemental Figure 5, left panel is confusing. The molecules of the two Copy1 and Copy2 should be colored distinctly. This creates a discrepancy between the text and supplementary figure for the interaction between UbD and RING domain for the crystal structure of R454Y RNF38.

The text says "Arg42 surface of UbD packs against the Tyr399, Gln407, and Glu434 regions of RNF38 in the asymmetric unit". But the supplementary figure 5 for the same shows interaction between UbD and RING of the same copy, which should ideally be between UbD and RING from two copies of the asymmetric unit.

4. Discrepancy in the figure 4f and 4g showing CSPs for XIAPWT and XIAP Y485R. The text claims "XIAP Y485R produced more signal attenuations in UbD and was the only variant to completely attenuate Gln40 of UbD". Though Gln40 was completely attenuated for Y485R variant compared to WT (figure 4h), CSP plots doesn't highlight more attenuation in Y485R compared to WT. Quantifying and plotting the peak intensity ratios (I/I₀) would have helped in better understanding the impact of Y485R mutation.

5. It is unclear why the E2 was precharged with Ub in the assay of Figure 1c. This is done primarily for single-round transfer assays, which is not the case here. I suggest the authors provide a sentence to explain the motivation.

RE: Life Science Alliance Manuscript #LSA-2025-03394-T

Thank you for submitting your manuscript entitled "Tuning ubiquitin transfer by RING E3 ubiquitin ligases through the linchpin residue" to Life Science Alliance. This manuscript was assessed by three expert reviewers, whose comments are below.

As you will see, reviewers were unanimous in their appreciation of the careful analysis of the lynchpin residue of this E3 ligase, with unexpectedly high ligase activity retained in some of the mutants. Reviewers 1 and 2 requested changes to the text to improve the discussion and rationale for some of the methods. Reviewer 3 was supportive but felt that some claims based on NMR analysis should be strengthened, for instance by expressing peak attenuation as a ratio of intensities. Given the overall strong support, we invite you to submit a revised manuscript that addresses all minor concerns, addresses the major concerns of Reviewer 3 in a manner of your choosing, and will not be subject to further peer review. We would be happy to publish your paper in Life Science Alliance pending these revisions as well as some changes necessary to meet our formatting guidelines.

We appreciate the quick response from the three reviewers, and are happy to hear the overall positive feedback. The changes to the text, formatting, and figures are addressed point-by-point below.

-Please be sure that the authorship listing and order is correct.

We have checked the author list and order with all co-authors, and the order appears correct. Importantly, the equal contribution from Mark A. Nakasone, Lori Buetow, and Mads Gabrielsen. In addition to Mark A. Nakasone and Danny T. Huang as co-corresponding authors.

To clarify the agreed order was:

Mark A. Nakasone^{1,5*}, Lori Buetow^{1,5}, Mads Gabrielsen^{1,4,5}, Syed F. Ahmed¹, Karolina A. Majorek¹, Gary J. Sibbet¹, Brian O. Smith², and Danny T. Huang^{1,3*}

-Please upload all figure files as individual ones, including the supplementary figure files; all figure legends should only appear in the main manuscript file.

We have high resolution files in illustrator format and have done this in your system.

-Please add a Running Title and a Summary Blurb/Alternate Abstract in our system.

Summary Blurb/Alternate Abstract : **"We demonstrate that altering a single residue in the RING domain greatly impacts ubiquitin transfer ability of RING E3 ligase; as a framework for verification, we combine functional, biophysical, structural, and cellular assays."**

Running Title: “**Linchpin residue governs activity of RING E3s**”

-Please add ORCID ID for secondary corresponding author - they should have received instructions on how to do so.

To clarify about the authors and ORCID, this list is below. It is CRUK policy to have our research integrity officer (Catherine Winchester c.winchester@crukscotlandinstitute.ac.uk) ensure this was complete prior to submission.

Mark A. Nakasone - <https://orcid.org/0000-0002-1362-191X>

mark.nakasone@glasgow.ac.uk

Lori Buetow - <https://orcid.org/0000-0003-4951-8057> |

l.buetow@crukscotlandinstitute.ac.uk

Mads Gabrielsen - <https://orcid.org/0000-0002-9848-2276>

Mads.gabrielsen@glasgow.ac.uk

Syed Feroj Ahmed - <https://orcid.org/0000-0003-1033-2538>

F.Syed2@crukscotlandinstitute.ac.uk

Karolina A. Majorek - <https://orcid.org/0000-0002-0973-9085>

kmajorek@gmail.com

Gary J. Sibbet – retired / doesn't have ORCID

Brian O. Smith – <https://orcid.org/0000-0003-3363-4168>

Brian.Smith@glasgow.ac.uk

Danny T. Huang - <https://orcid.org/0000-0002-6192-259X>

d.huang@crukscotlandinstitute.ac.uk

-Please add a Category for your manuscript in our system.

The category is Biochemistry and Chemical Biology; Structural Biology and Molecular Biophysics

-Please add the X and Bluesky handles of your host institute/organization as well as your own or/and one of the authors in our system.

We mainly use LinkedIn, Twitter/X, and some use BlueSky and Research gate.

CRUK Scotland Institute (formerly CRUK Beatson)

https://x.com/CRUK_SI

<https://www.linkedin.com/company/cancer-research-uk>

Mark A. Nakasone - https://x.com/Mark_A_Nakasone

<https://www.linkedin.com/in/mark-a-nakasone-ph-d-b9747354/>

Lori Buetow - <https://www.linkedin.com/in/lori-buetow-8b49981/>

Mads Gabrielsen - <https://bsky.app/profile/mgabro.bsky.social>
<https://www.linkedin.com/in/mads-gabrielsen-255a6755/>

Syed Feroj Ahmed - <https://www.linkedin.com/in/syed-feroj-ahmed-bb20741b/>
<https://x.com/ferojahmed>

Karolina A. Majorek - <https://www.linkedin.com/in/karolina-a-majorek-nakasone-phd-0b071a219/> (No Twitter/X or BlueSky)

Gary J. Sibbet – <https://x.com/gazsib> (No LinkedIn)

Brian O. Smith – <https://www.linkedin.com/in/brian-smith-43802444/>
<https://bsky.app/profile/brianosmith.bsky.social>

Danny T. Huang - <https://www.linkedin.com/in/danny-huang-857399/>
(No Twitter/X or BlueSky)

-Please be sure that the authorship order is correct and matches between the system and the manuscript file.

We have double checked this.

-Please rename your sections according to our guidelines and be sure to include all of them as listed here: Title page, Summary blurb, Abstract, Introduction, Results, Discussion, Materials and Methods, Data Availability, Acknowledgements, Author contributions, Conflict of interest, References, Figure legends, including supplementary figure legends.

We have included this in the submitted version.

-Please add an Author Contributions section to the system as well.

We have included this in the submitted version, it can be extended, but captures the unique contributions of all authors.

1. Mark A. Nakasone

Cancer Research UK Scotland Institute, Gartnavel Estate, Switchback Road, Glasgow, G61 1BD, United Kingdom

Current:

Contribution

Conceptualization, Methodology, Validation, Data curation, Formal analysis, Investigation, Visualization, Resources, Writing - original draft, Writing – Review & Editing

Contributed equally with

Lori Buetow, Mads Gabrielsen

Competing interests

No competing interests declared

2. Lori Buetow

Cancer Research UK Scotland Institute, Gartcubie Estate, Switchback Road, Glasgow, G61 1BD, United Kingdom

Contribution

Conceptualization, Methodology, Validation, Formal analysis, Investigation, Visualization, Resources, Writing – Review & Editing

Contributed equally with

Mark A. Nakasone, Mads Gabrielsen

Competing interests

No competing interests declared

3. Mads Gabrielsen

Cancer Research UK Scotland Institute, Gartcubie Estate, Switchback Road, Glasgow, G61 1BD, United Kingdom

Current: Neil Bulleid Integrated Protein Analysis, University of Glasgow, Glasgow, G12 8QQ, United Kingdom

Contribution

Conceptualization, Methodology, Validation, Data curation, Formal analysis, Investigation, Visualization, Resources, Writing – Review & Editing

Competing interests

No competing interests declared

4. Syed F. Ahmed

Cancer Research UK Scotland Institute, Gartcubie Estate, Switchback Road, Glasgow, G61 1BD, United Kingdom

Contribution

Methodology, Validation, Investigation, Resources, Writing – Review & Editing

Competing interests

No competing interests declared

5. Karolina A. Majorek

Cancer Research UK Scotland Institute, Gartcubie Estate, Switchback Road, Glasgow, G61 1BD, United Kingdom

Contribution

Validation, Investigation, Resources, Writing – Review & Editing

Competing interests

No competing interests declared

6. Gary J. Sibbet

Cancer Research UK Scotland Institute, Gartcubie Estate, Switchback Road, Glasgow, G61 1BD, United Kingdom

Contribution

Investigation, Formal analysis

Competing interests

No competing interests declared

7. Brian O. Smith

School of Molecular Biosciences, University of Glasgow, Glasgow G12 8QQ, United Kingdom

Contribution

Investigation, Formal analysis, Writing - Review & Editing

Competing interests

No competing interests declared

8. Danny T. Huang

Cancer Research UK Scotland Institute, Gartcubie Estate, Switchback Road, Glasgow, G61 1BD, United Kingdom

School of Cancer Sciences, University of Glasgow, Glasgow, G61 1BD, United Kingdom

Contribution

Conceptualization, Resources, Funding acquisition, Writing – Review & Editing

Competing interests

Consultant for Triana Biomedicines

For correspondence

Danny T. Huang and Mark A. Nakasone

Competing interests

No competing interests declared.

-Please add your main, supplementary figure, and table legends to the main manuscript text after the references section.

We have added these captions/legends in our updated manuscript.

-Please label the panels in Figure S3 accordingly.

We have labelled Supplementary Figure 3 with “(a)” and “(b)”, also added “MW (kDa) for the panel b. Should be more clear these are the same gels visualized two different ways.

-We encourage you to revise the figure legend for Figure S8 such that the figure panels are introduced in alphabetical order.

We have done this in line with the manuscript, the standard load for Supplementary Figure 8a is introduced first, then the full ^1H , ^{15}N -HSQC NMR spectra of UBE2D2^{S22R,C85K}-Ub(^{15}N) with XIAP (WT, Y485R, and Y485E) corresponding to Supplementary Figure b,c,d. In the legend it is now clear what a, b, c, and d are for Supplementary Figure 8.

-Please add callouts for Figures 4i; S3A-B; S6A-C and S7A-D to your main manuscript text.

There is now a new sentence to introduce the concept for Figure 4i. In the context of supplementary figure 3 a&b, we are refereeing to the whole figure (two visualizations of the same gel). The callout for S6a and then S6b&c are now in the manuscript, as well as 7a-d.

According to our institute’s policy, CRUK’s Press Office guidelines:

“CRUK wants to maximise the impact of research they fund and require their funded researchers to inform them when a paper is submitted or an abstract is accepted at a conference. The Head of the Research Integrity Service will provide information on submitted papers to CRUK’s media team on a regular basis. Notification can also be made via the online manuscript submission form available on the CRUK website:

<https://www.cancerresearchuk.org/manuscript-submission>

Or you can email the media team directly:

Graeme.Sneddon@cancer.org.uk

Ana.Barros@cancer.org.uk

Phil.Prime@cancer.org.uk

fiona.macleod@cancer.org.uk

*Please contact the Head of the Research Integrity Service for any further information or help with any of the above.

c.winchester@crukscotlandinstitute.ac.uk

In line with our policy, we can cc LSA to find a realistic date for the press release.

LSA now encourages authors to provide a 30-60 second video where the study is briefly explained. We will use these videos on social media to promote the published paper and the presenting author (for examples, see <https://docs.google.com/document/d/1-UWCfbE4pGcDdcgzcmiuJl2XMBJnxKYeqRvLLrLSo8s/edit?usp=sharing>). Corresponding or first-authors are welcome to submit the video. Please submit only one video per manuscript. The video can be emailed to contact@life-science-alliance.org

We have discussed this short video with first and corresponding authors. Like your examples, we have highlighted the major findings in PowerPoint format with all co-authors for the video.

To upload the final version of your manuscript, please log in to your account: <https://lsa.msubmit.net/cgi-bin/main.plex>

We have used the link to submit the final version of the manuscript.

A. FINAL FILES:

We have kept MS word format with EndNote formatting for references.

We have decided on the following summary blurb, “**We demonstrate that altering a single residue in the RING domain greatly impacts ubiquitin transfer ability of RING E3 ligase; as a framework for verification, we combine functional, biophysical, structural, and cellular assays.**”

B. MANUSCRIPT ORGANIZATION AND FORMATTING:

Where possible, the source data is included, however large data sets such as the raw reflections from the crystal structures are in the open data repository under the “Data Availability” link (Zenodo) and other data is in the protein data bank (PDB). In addition, the same copy of raw source data is backed up at CRUK-SI as per our research integrity policy. This same archive was upload to Zenodo to increase accessibility and transparency. To the best of our ability, full gel images (Western Blots, In-gel fluorescence, and Coomassie) are included in the supplements and source data. The open data will have the raw image formats as they were recorded on digital scanners (e.g. Epson 750 scanner, BioRad ChemiDoc MP, LiCor CLx).

We checked the Zenodo ([10.5281/zenodo.15178736](https://doi.org/10.5281/zenodo.15178736)) for our raw data, and all is included.

We have received this link in a separate email and provided the signature. Our research integrity officer has gone over relevant funding/awards/grants. All authors are CRUK-SI (Beatson) or University of Glasgow employees for this study. N/A for US/NIH or UK/Crown employment.

We do not have any strong feelings against this transparent peer review process.

As we received this at 19:00 on Friday (GMT/UTC+00:00), we may need a few more days.
*Thank you for the extension confirmed in a previous email.

Sincerely,

Reviewer #1 (Comments to the Authors (Required)):

The manuscript describes the importance of the linchpin arginine (LP-Arg) residues on loop two on RING-type E3 ligases using the RNF38 RING E3 ligase protein. To perform these studies, the group mutate the LP-Arg to the 19-other amino acids and test their activity via autoubiquitination and perform NMR studies as well as co-crystallization with several mutants. The for various mutant more polyubiquitinated product, the more active the enzyme. Ultimately, the result of this extensive work confirms the importance of LP-Arg in Ub chain elongation by RING E3 ligases. To perform some of these studies, they solved the first structure of the RING domain of RNF38. In addition, the solved the structure of RING domain in complexed with the E2 enzyme covalently attached to the active site Ub. Lastly, the use the XIAP RING protein with weak activity and engineer a RING - protein with the binding domain of XIAP substrate and demonstrate that they can polyubiquitinate that protein. They furthermore introduced the LP0-Arg in XIAP which lacks an arginine and demonstrate substrate ubiquitination by XIAP-Arg.

As a whole, these is an extensive body of work that confirms the importance of the H-bond between a glutamine adjacent to the catalytic cysteine on the E2 and this LP-Arg on loop2. However, its work publishing but hopefully with minor important that should highlight the extensive work of the manuscript

I would have like to see greater discussion on the structure of the RING domain, as RING domains are not common, purification conditions of wt and mutants, and how its

structure may be different or similar to other RING and the location of the LP-Arg. I would have also liked greater discussion on the complexes unless this is repeated work of others and a better and confirm others works, which then takes away from the importance; thus the authors should highlight novel finding. If the authors chose not to highlight some of this exciting work, then it would reflect confirming work of several published work.

In conclusion, I don't have an major criticism of this manuscript. Its well written, contains a lot of data and effort that I think needs described and highlight in the manuscript and not in supplemental material. As presented, its seems to target a very narrow and mall subset of ubiquitination community.

We thank Reviewer 1 for the accurate summary of our manuscript and request for clarification on our experimental methods. Overall, the reviewer is correct that this study is highly focused on a single aspect of an evolutionally conserved feature of RING domains, the cationic “linchpin residue.” This biochemical detail has broad implications for post-translational modifications (PTMs) and ubiquitin transfer, as RING-type E3 ligases are the most common in humans, numbering ~634.

To address the reviewer’s points, we have better highlighted the importance of RING domains both in the introduction and discussion sections. Furthermore, the reviewer is correct in that many of these RNF38 mutants have not been reported and to date no study has systematically investigated them in combined functional, binding, and structural assays. Referring to our methods sections, we note that we generated the RNF38-RING linchpin point mutations with conventional site directed mutagenesis (SDM) PCR. Also, the *E. coli* expression method (T7, IPTG, and ZnSO₄ supplemented to 200uM) of the GST-TEV fusion variants produced soluble well-folded RING mutants. We have expanded the raw data (Zendo) to include SDM primers and clarified expression conditions in the methods. Furthermore, in the near future we will deposit select *E. coli* expression constructs to AddGene to make them more accessible.

Many E2-Ub/RING complexes have been reported and are analysed in Supplement 1b-g, but thus far the field has neglected the linchpin mutations. Several studies have introduced the most disruptive linchpin mutations in functional assays, but not explored the effects of less disruptive mutations nor provided any explanation about their structural and functional outcomes. Bioinformatics studies have analysed human RING domains but lack experimental validation. We have altered figure 1 to highlight this and left Supplementary Figure 1 that summarizes decades of work on RING E3s.

The novelty of this study lies in how we can manipulate a single residue to influence ubiquitin transfer. This is demonstrated *in vitro* and in cells, with a clear distinction between E2~Ub binding affinity and ubiquitin ligase activity.

We have expanded the language to reflect this for the PTM, ubiquitin, and protein design fields. If current trends prevail, de novo protein design with generative methods could introduce new RING-based E3s that will need to account for optimizing RING domain function, in which the identity of the LP plays a big part.

Reviewer #2 (Comments to the Authors (Required)):

Interesting study that looks at the importance of and the chemical nature of the amino acid on the linchpin position in RING E3-ligases. It is to be expected that deviation from Arg to anything else (RNF38) will result in a drop of affinity and potentially also activity; strikingly this is not the case for all mutations; and a molecular explanation is given based on NMR-experiments for one deviating mutant. In addition, work on a XIAP-RNF38 chimera reveals that when replacing the native Tyr with an Arg an increase in activity is seen. Also, activity in vivo is studied, to start casting a light on the cell biological importance of LPs. Based on my evaluation the manuscript could be well suited for publication after some minor clarifications have been addressed. The study is a bit fragmented as studies on multiple ligases (RNF38 and XIAP-RNF38 chimera) is presented, also multiple cell lines are used for in vivo work.

We thank the reviewer for the accurate interpretation of our study and experimental approaches. Often, studies focus on a single RING E3 or a related family of RING E3s. We thoroughly investigated the impact of the identity of the linchpin residue in RNF38 across functional, binding, and structural assays; however, one setback for cell-based studies was the lack of known biology for RNF38. By introducing the XIAP-RNF38 chimera, we could demonstrate a successful approach to protein design and highlight the principal that the identity of the linchpin residue truly has major implications for ubiquitin transfer. In addition, XIAP was an obvious candidate for linchpin manipulation because the wild type form is divergent in this position (LP^{Tyr}). The cell lines were chosen based on the basal expression of substrate, E3, and active pathways for both. The discussion has been slightly altered to reflect these intentions and points. Taken together, we combined these diverse experimental methods to report the whole story of how this single linchpin residue impacts E3 ligase activity.

-Can the authors give a rationale why initially XIAP induced SMAC Ubiquitination is monitored and on a later stage XIAP induced PTEN ubiquitination?

The use of multiple substrates (SMAC and PTEN) for the in vitro and then cell-based assays, proves that the gained activity of the E3 ligase XIAP is generalizable across various substrates tested and not specific to a particular substrate. The use of XIAP/SMAC for in vitro assays is mainly due to the properties of these proteins. Mature SMAC (56-C) is expressed well from *E. coli*, whereas obtaining PTEN is challenging, even from eukaryotic systems, and the protein is much less stable. Furthermore, decades of work report numerous binding partners for XIAP including PTEN, which is a reliable target in cells that is easily monitored by well-established primary antibodies. Primary antibodies for SMAC detect multiple forms (mature and precursor) making Western blot detection challenging – XIAP does not ubiquitinate the precursor of SMAC, which lacks an N-terminal Ala-Val-Pro-Ile motif to bind XIAP's BIR domain.

-Can the authors give a rationale for why initially HEK293s are used and in the chx

experiment U2Os are used?

Similarly to above, we used two different cell lines (HEK293 and U2OS) as a common practice to test the robustness of our findings in different cellular settings and show that the results are not centred around the unique characteristics of a single cell line.

Reviewer #3 (Comments to the Authors (Required)):

Comments on Tuning ubiquitin transfer by RING E3 ubiquitin ligases through the linchpin residue by Nakasone et al

RING E3s represent a major class of E3s that rely on their catalytic RING domain to interact with the E2~Ub conjugate. A key feature of this domain is a conserved arginine residue, often referred to as the 'linchpin arginine,' which stabilizes the E2~Ub in a closed, catalytically competent conformation, thus promoting efficient ubiquitin transfer.

This manuscript investigates the active role of linchpin residue in mediating ubiquitin transfer and shows how different residues at this position modulate ubiquitination efficiency to varying degrees by differentially stabilizing the E2~Ub closed conformation. The authors suggest a combination of binding and NMR experiments to evaluate activation of E2 by RING-E3s.

The linchpin residue has been essential for understanding the activity of ubiquitin E2/E3 enzymes and has broad implications for protein regulation. The findings of the manuscript will be of interest to the community. I have a few concerns regarding the interpretation of the data.

This is a concise summary from the reviewer regarding our study. This review suggests the focus on the linchpin residue (LP) in RING E3s and the approach to decouple RING/E2~Ub binding, donor Ub conformation in E2~Ub, and activity came across as important concepts to the reviewer. We are pleased that these distinct aspects of RING E3s with E2~Ub are clear to the reviewer.

Major Comments

1) Linchpin arginine establishes an intricate Hydrogen bond network with E2 and Ub to stabilize the closed conformation, where arginine acts as an H-donor. In the paper, substitutions like methionine, a non-H-bonding residue, support higher ubiquitination efficiency compared to serine or tyrosine, which are capable of H-bonding. This confuses the mechanistic insight into how LP substitution modulates activity. The NMR experimental CSPs of E2~Ub(15N)/RNF38-R454M complex and E2~Ub(15N)/RNF38-R454S may shed some light.

The editor has given 7 days for revisions and these NMR experiments may not be feasible. We do note that RNF38 with LP mutations Arg-to-Ala have been reported to nearly abolish all E3 ligase activity with another E2, UBE2K (<http://doi.org/10.1038/s41589-021-00952-x> figure 2C). We need to clarify that while LP Arg stabilizes E2-Ub and RNF38 junction via network of hydrogen bonds, this does not imply that hydrogen bond interaction is the sole criteria to stabilize this junction to

maintain the closed conformation. The guanidinium group of LP Arg also stacks against Leu71 sidechain from donor Ub. Notably, no other amino acid can mimic the interaction network established by Arg, as we see in Figure 1E (activity) and Table 1 (SPR E2~Ub binding). It is likely that different amino acids at this position engage distinct interaction, resulting in varying capacities to stabilize the closed E2-Ub conformation and hence activity as revealed by our NMR data. For example, a prior structural study of the TRAF6-E2-Ub complex (PDB: 5VNZ) showed that the LP Asn sidechain forms hydrogen bond only with Ub's Gln40 sidechain. Ser is short and modelling suggests that Ser could form hydrogen bond with Ub's Gln40 sidechain. In contrast, Met could form hydrophobic interaction with Ub's Leu71. Even LP Lys and Met had deficiencies in our study, suggesting LP Arg is optimal. Last, we utilized RF diffusion based structural prediction (AlphaFold3 and Chia1) to predict the structure of all 20 LP residues with E2~Ub and these returned essentially the same structure. This highlights the need for our study and that we have not reached a point computationally where we can reliably predict these subtle, but important interactions. *please see attached PowerPoint file, also in raw data at Zenodo ([10.5281/zenodo.15178736](https://doi.org/10.5281/zenodo.15178736)).

2) I have a few concerns regarding the discussion of the NMR assay and data in the manuscript, which are listed below.

i) Despite the E2~Ub/WT-RNF38 adopting a single closed conformation, why do some peaks broaden while others do not? This point remains unclear and has not been addressed.

The 2D peaks represented in the $^1\text{H}, ^{15}\text{N}$ -HSQC spectra reflect a unique J-coupled ^{15}N - ^1H pair. In the top right of the spectra, some of these NH groups are from side chains, but many of the well spread ones would be from amides of unique residues (excluding proline). This spectra of ^{15}N -Ub^D on E2-Ub, we denote "E2-Ub(^{15}N)" to communicate what is ^{15}N labelled. In general, the spectra of ubiquitin (Ub) is well known and has been assigned numerous times. Upon titration, cross-peaks in $^1\text{H}, ^{15}\text{N}$ -HSQC spectra can broaden (decrease intensity) for some common reasons: 1) the binding putting the labelled entity into the slow exchange regime, 2) the labelled entity experiencing a large increase in molecular weight (increase in T1 relaxation), or 3) paramagnetic or pseudocontact shifts from the binding partner. There are no unpaired electrons or metal ions aside from Zn^{2+} in RNF38 in the NMR samples so option 3 is very unlikely. Case 2 could be possible, but while loaded to the active site of E2, Ub(^{15}N) in E2-Ub(^{15}N) is extremely flexible. The RING domains added are small, the 79 residue RNF38 RING domain we used is 9.1 kDa, the E2 is ~17kDa. All together formation of the E2-Ub(^{15}N)/RNF38 complex (1:1:1) would only be ~35 kDa. Based on the experiment of E2-Ub(^{15}N)/RNF38^{R454E} and related mutants, we begin to rule out increase in size. Therefore, the reason 2D peaks broaden in the spectra is due to Ub(^{15}N) entering the slow exchange regime with RNF38 wild type, but not disruptive LP mutants. This is consistent with the strong binding of RNF38 to E2-Ub. The remaining peaks are not as involved in binding as the attenuated peaks. We note this concept opens deeper discussion on microscopic (residues specific) vs. global parameters that NMR can deliver.

We have included wording in the manuscript's results section, supplementary figure legend, and discussion to clarify this. For the journals word count and references we had to omit much of the history and detailed explanations regarding applying NMR to protein-protein interactions.

ii) It is better to present peak attenuation as a ratio of intensities (I/I₀) to better understand the data. This would clarify a few ambiguous texts in the document, such as "XIAP Y485R produced more signal attenuations in UbD and was the only variant to completely attenuate Gln40 of UbD."

In general, this would be an excellent idea to present the ratio of intensities (I/I₀). In practice, residue specific T1 or T2 relaxation would be more accurate for comparing between mutants and samples. However, with the numerous signal attenuations in the wild type RNF38 or LP-Arg XIAP (Y485R), we set a common threshold/contour level to display the spectra. Any peak not observable was classified as attenuated, and we further checked manually going through each spectrum with different contour thresholds starting in the noise and slowly increasing. In many cases, the attenuated peaks were completely gone at 1:1 (E2-Ub:RING). In this case when we apply a peak model to integrate these broadened peaks to get a volume or attempt to just measure intensity, it is effectively zero - the same level of the noise. With the current spectra and nature of binding (slow exchange regime) the proposed I/I₀ analysis would lead to a confusing plot that is essence the inverse of our current CSPs + signal attenuation plots.

We note that presenting the signal attenuations with the CSP bar plots provides a more complete picture of binding as some residues did not attenuate but did exhibit change in chemical shift perturbation (CSPs). Last, we do have a single intermediate titration point at 0.5 before reaching 1:1, but with signal attenuation with LP-Arg RING is not so gradual when going to the slow exchange regime. We have expanded on how the NMR data was displayed in the experimental methods section and figure legend, including attempting more scans (ns=256) to recover the attenuated peaks – which did not work. In addition to including all raw NMR data on XIAP at Zenodo ([10.5281/zenodo.15178736](https://zenodo.org/record/10.5281/zenodo.15178736)).

iii) The number of attenuated signals has been repeatedly used in the text as a sensitive measure of activation and closed conformation. In contrast, it is a very poor measurement that provides no quantification. I strongly suggest removing such statements from the text. Or the authors need to provide a quantification of closed/open conformations.

The reviewer is referring to the ¹H,¹⁵N-HSQC NMR spectra comparing the beginning (no RING added) to the end point. Presenting the full ¹H,¹⁵N-HSQC NMR spectra and is a common qualitative means to demonstrate clear differences in binding. In our experimental design, the RING domain either: 1) minimally interacts with E2-Ub resulting in minimal change the spectra, 2) an intermediate LP variant RING binds but does not produce the large-scale attenuations as wild type / LP-Arg, or 3) The RING has an optimal LP residue (RNF38 wild type or XIAP-YtoR), in which case nearly all signals

are lost – with clear indicators of slow exchange. We never intended for this to be a quantitative measurement and have changed to wording to reflect this. We do feel it is important to present the full NMR spectra and allow readers to qualitatively compare them. As discussed above, quantitative analysis of such attenuated spectra is problematic and potentially misleading. The more expensive ^{13}C methods likely would be better suited for such analysis and we have included this in the discussion. Such experiments would go beyond the current scope of our study, but certainly promising for more detailed NMR studies on RING E3 systems. We tried analysis of intensity several ways, with the most promising approach being normalizing Q40 to a common peak that is unaffected by binding serving as internal standard. *please see the analysis in the attached PowerPoint file, included in raw data at Zenodo ([10.5281/zenodo.15178736](https://zenodo.org/record/10.5281/zenodo.15178736)).

Minor Comments

1. Some of the subheadings for "results" section can be reconsidered to convey the key findings of the experiment. For example, "Effect of LP variants on ubiquitination" can be rephrased to "LP variants differentially modulate ubiquitination efficiency".

Other subheadings that can be rephrased include:

- "Binding of RNF38 LP mutants to UBE2D2 and UBE2D2-Ub"
- "Structure of E2-Ub with RNF38 RING LP variants"
- "Effect of LP-activated XIAP in cells"

According to the journal, there are clear guidelines for the length of the results heading, we attempted to meet both for the reviewer and journal. We believe the reviewer's suggestions are an improvement to convey the findings.

Results sections are now re-worded to:

LP variants differentially modulate ubiquitination efficiency

Structure of E2-Ub with RNF38^{RING} LP variants perturbs donor Ub

RNF38 LP mutants alter binding to UBE2D2 and UBE2D2-Ub

2. There is a discrepancy in the statement "The deficiency of the R454Y variant in the ubiquitination assay compared to the R454K, R454M, and R454W variants was unexpected as it exhibited the second-best binding affinity to UBE2D2S22R, C85K-Ub after RNF38RING-WT".

According to table 1 depicting affinities of E2 and E2-Ub conjugate, both R454W and R454Y have similar affinities for E2-Ub conjugate.

In regard to LP-Arg (wt), LP-Lys, LP-Met, LP-Tyr, and LP-Trp, they were subjected to activity assay and binding to E2/E2~Ub in SPR. For the activity assay (Figure 1C&E) “Fraction Ub depleted” the highest was LP-Arg (~0.3), while LP-Lys and LP-Met (~0.11), then LP-Trp (~0.85), and then LP-Tyr was nearly inactive.

Activity assay Ub depletion: LP-Arg>>LP-Lys=LP-Met> LP-Trp>>LP-Tyr

Binding E2~Ub SPR (Kd uM): LP-Arg (2.5 ± 0.4 uM)>>LP-Tyr (23.8 ± 1.1 uM)=LP-Trp (24.3 ± 0.8 uM)>LP-Lys (30.2 ± 3.7 uM)=LP-Met (31.2 ± 1.7 uM)

The use of “deficiency” refers to measurements from two different experiments monitoring different properties – ubiquitin transfer and binding. One of our key messages is that there is a “discrepancy” between these assays that was unpredictable. As an example LP^{Lys} has the second best Ub depletion, but binds similar to other LP mutants that were more deficient in Ub depletion. The fact that LP mutants do not directly correlate between Ub depletion and SPR binding prompted us to continue investigation. This adds to the novelty of our findings and experimental approach, as neither activity or binding alone can be used to predict the impact on RING mutations. Indeed, R454W and R454Y have near identical binding to E2-Ub in SPR, but Ub depletion (activity) are notably different (Figure 1E). Therefore, when assessing RING E3s or mutations, it is important to decouple E2-Ub binding from activity. Application of our NMR approach provides insight on how donor Ub behaves and although challenging, X-ray crystallography can be used to determine the whole structure.

3. Supplemental Figure 5, left panel is confusing. The molecules of the two Copy1 and Copy2 should be colored distinctly. This creates a discrepancy between the text and supplementary figure for the interaction between UbD and RING domain for the crystal structure of R454Y RNF38.

The text says "Arg42 surface of UbD packs against the Tyr399, Gln407, and Glu434 regions of RNF38 in the asymmetric unit".

But the supplementary figure 5 for the same shows interaction between UbD and RING of the same copy, which should ideally be between UbD and RING from two copies of the asymmetric unit.

There are two colors, we now have colored each complex with similar but different colors to make a distinction between the two copies in Supplementary Figure 5. These are two copies in the same asymmetric unit, we continue to Supplement Figure 6 to rule out Ub binding since in the crystal it is unclear if this interaction is real or an artefact of crystal packing that we observed through symmetry contacts. In Supplementary Figure 5b, only one copy of UBE2D2^{S22R,C85K}-Ub/RNF38^{R454Y} is shown for the comparison with PDB-4V3K. Note this structure also has two copies of the complex in the asymmetric unit.

4. Discrepancy in the figure 4f and 4g showing CSPs for XIAPWT and XIAP Y485R. The text claims "XIAP Y485R produced more signal attenuations in UbD and was the only variant to completely attenuate Gln40 of UbD". Though Gln40 was completely

attenuated for Y485R variant compared to WT (figure 4h), CSP plots doesn't highlight more attenuation in Y485R compared to WT. Quantifying and plotting the peak intensity ratios (I/I₀) would have helped in better understanding the impact of Y485R mutation.

We have reassessed the spectra including those not included. At the 1:1 ratio, XIAP(Y485R) abolished all signal from Q40 in E2-Ub(15N), even with 256 scans (ns=256). It is true that XIAP^{WT} has a similar effect, but notably in figure 4h, Q40 from donor Ub is split in at least two signals (state 1,2, and3), while Q40 with XIAP^{Y485E} overlays much better as a single peak with E2-Ub(15N) alone (no RING). In general, the presence of a peak in 2 or more states makes this analysis challenging. We did try to recover this information by increasing the number of scans (ns) in the 2D 1H,15N-HSQC spectra. This is now clear in the methods and revised figures & captions. Additionally, we have also included these with our raw data (Zendo link). Notably, even with ns=256 at the 1:1 points, we still could not recover the Q40 signal in E2-Ub(15N) for XIAP^{Y485R} – supporting our point about strong binding. As the split signals for Q40 are still observable, XIAP^{WT} is inferred to be in two states. The nature of these titration experiments (adding XIAP-RING to the NMR tube) would change the concentration making I/I₀ less accurate, but we did attempt to account for the concentration, receiver gain (rg), spectral noise, and number of scans (ns). We doubled checked 0 order and 1st order phasing, used the same optimized window function, and ensured linear prediction did not introduce artefacts.

We tried analysis of intensity several ways, with the most promising approach being normalizing Q40 to a common peak that is unaffected by binding serving as internal standard. *please see the analysis in the attached PowerPoint file (also in raw data).

5. It is unclear why the E2 was precharged with Ub in the assay of Figure 1c. This is done primarily for single-round transfer assays, which is not the case here. I suggest the authors provide a sentence to explain the motivation.

This is a good point regarding precharging E2 with Ub. The precharged assays were started from the same point, rather than quenching the reaction (adding EDTA) and allowing for single-turnover (as described <https://pubmed.ncbi.nlm.nih.gov/30242700/>). Single-turnover assays were not feasible for this study. In the absence of continuous charging, E2~Ub dissociated more rapidly than ubiquitin transfer occurred for the slower mutants, making it impossible to compare ligase activity. The primary reason for the difference in precharging for the autoubiquitination assays and not precharging for the substrate ubiquitination assays relates to measurement sensitivity when measuring a decrease in signal (use of Ub in the autoubiquitination assays) versus an increase in signal with a starting point of 0 (the appearance of ubiquitinated substrate). Signal differences of 10% or less are indistinguishable when assessing band intensity reduction using labeled ubiquitin. This signal measurement variability increases when multiple small volume pipette transfers are used to set up individual reactions for each RNF38 variant. By precharging, we used the same charge stock reaction across all RNF38 variants, allowing for higher pipette volumes and minimizing the number of pipette transfers per reaction.

July 8, 2025

RE: Life Science Alliance Manuscript #LSA-2025-03394-TR

Prof. Danny T. Huang
Cancer Research UK Scotland Institute
Cancer Research UK Scotland Institute, Garscube Estate, Switchback Road, Glasgow, G61 1BD, United Kingdom
Garscube Estate
Switchback Road
Glasgow, Scotland G61 1BD
United Kingdom

Dear Dr. Huang and Dr. Nakasone,

Thank you for submitting your revised manuscript entitled "Tuning ubiquitin transfer by RING E3 ubiquitin ligases through the linchpin residue". We appreciate the diligence with which you have approached the suggestions from Reviewer 3. The editors do not feel these additional data provided in response to this reviewer must be included as supplementary figures in the manuscript. However we remarked that these are included in your files uploaded to zenodo. If you wish, you may include references to data held in this repository in the text of the manuscript to support interpretation on the linchpin residue by AlphaFold3 modeling and to offer greater context on your NMR spectroscopy analysis.

We leave these text changes, and references to the data on zenodo, to your discretion. In the meantime please attend to the remaining points:

- Please provide clean manuscript file, without tracked-changes.
- For publication, we require PowerPoint, TIFF, PDF or EPS figure files.
- Please add ORCID ID for the corresponding author Huang--you should have received instructions on how to do so.
- Please use the [10 author names, et al.] format in your references (i.e. limit the author names to the first 10).
- After the 'References' section please place Figure Legends then Supplementary Figure Legends and then at the end, Table 1 and 2.
- Please add callouts for Supplemental Figure 3 panels A and B and Supplemental Figure 5 panels A and B to your main manuscript text.

A. FINAL FILES:

B. MANUSCRIPT ORGANIZATION AND FORMATTING:

Sincerely,

July 14, 2025

RE: Life Science Alliance Manuscript #LSA-2025-03394-TRR

Prof. Danny T Huang
Cancer Research UK Beatson Institute
Garscube Estate, Switchback Road
Glasgow, Scotland G61 1BD
United Kingdom

Dear Dr. Huang,

Thank you for submitting your Research Article entitled "Tuning ubiquitin transfer by RING E3 ubiquitin ligases through the linchpin residue". It is a pleasure to let you know that your manuscript is now accepted for publication in Life Science Alliance. Congratulations on this interesting work! Thank you for your careful attention to reviewer comments and the video summarizing your findings.

DISTRIBUTION OF MATERIALS:

Again, congratulations on a very nice paper. I hope you found the review process to be constructive and are pleased with how the manuscript was handled editorially. We look forward to future exciting submissions from your lab.

Sincerely,
